# Rectifying Adaptive Learning Rate Variance via Confidence Estimation

## Abstract

Recent advances in training physics-informed neural networks (PINNs) highlight the effectiveness of second-order optimization methods. Adaptive variants such as AdaHessian, Sophia, and SOAP leverage approximate curvature information to achieve strong performance on challenging benchmarks. However, adaptive optimizers are prone to instability during the early stages of training—a limitation addressed in part by RAdam through rectification of the adaptive learning rate. We introduce Adaptive Confidence Rectification (ACR), a novel uncertainty-aware rescaling mechanism that enhances RAdam's rectification strategy by dynamically adjusting the learning-rate correction based on an empirical measure of confidence. Our method integrates seamlessly with diverse optimizers and training regimes, consistently improving convergence stability and optimization accuracy. Extensive experiments on large-scale PINN tasks demonstrate reliable performance gains over both rectified and non-rectified baselines, establishing ACR as a robust and broadly applicable optimization framework.

## 1 Introduction

Second-order optimization has emerged as a cornerstone in training Physics-Informed Neural Networks (PINNs) (Raissi et al., 2019), offering substantial advantages over first-order methods. By incorporating curvature information, second-order algorithms better align optimization dynamics with the functional geometry of the loss landscape, enabling faster convergence and improved stability—particularly in stiff, high-dimensional, and multi-scale PDE settings such as fluid dynamics, structural mechanics, and power systems (Müller & Zeinhofer, 2024; Rathore et al., 2024; Bonfanti et al., 2024; Cai et al., 2021; Mao et al., 2020; Misyris et al., 2020).

Recent adaptive second-order methods, including SOAP (Vyas et al., 2025), AdaHessian (Yao et al., 2021), and Sophia (Liu et al., 2024a), combine Hessian approximations with exponential moving averages (EMAs) to stabilize both gradient and curvature estimates. Among these, SOAP has drawn particular attention for achieving state-of-the-art performance across diverse PDE benchmarks (Wang et al., 2025), outperforming momentum-based optimizers such as Adam (Kingma & Ba, 2015).

Despite these advances, adaptive optimizers remain vulnerable to instability in early training, primarily due to variance inflation in the second-moment EMA (Liu et al., 2020). Variance rectification techniques—most notably RAdam (Liu et al., 2020)—mitigate this issue by adaptively normalizing EMA variance during initial iterations, a strategy now widely adopted across machine learning (Song et al., 2023; Luo et al., 2023; He et al., 2020; Katharopoulos et al., 2020; Yasunaga et al., 2021). The importance of rectification is particularly pronounced in PINNs: Figure 1 (left) shows how gradient distributions evolve from bimodal to unimodal within the first 25 epochs, indicating unstable variance, while Figure 2 (left) highlights SOAP's oscillatory behavior and slow convergence on the Rosenbrock function—a canonical test of optimizer stability.

However, RAdam's effectiveness is fundamentally limited by its reliance on the assumption that gradients follow an initial zero-mean Gaussian distribution. In practice, gradients in deep networks are often heavy-tailed or multimodal, undermining this theoretical basis. Specifically, RAdam's rectification depends on the effective length of the moving average, $\rho_t$, which is derived under restrictive distributional assumptions. This design leads to three key limitations: (1) the Gaussian assumption is frequently violated, invalidating the variance estimate; (2) the fixed target variance is

Figure 1: Comparison of histograms for log scale absolute gradients over the first 25 training iterations of a PINN applied to the Navier-Stokes problem. SOAP (left) exhibits significant gradient distortion, while the rectified version avoid distortion: both RAdam (center) and our method ACR (right) preserves the bimodal Gaussian structure.

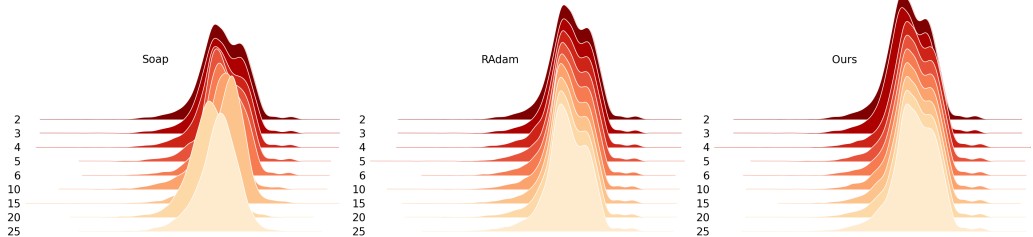

Figure 2: Optimization performance on the Rosenbrock function. SOAP exhibits slow and oscillatory convergence across a range of $\beta_1$ values, highlighting instability during early training. In contrast, both RAdam and ACR achieve significantly faster and smoother convergence. All runs use $\beta_2 = 0.999$.

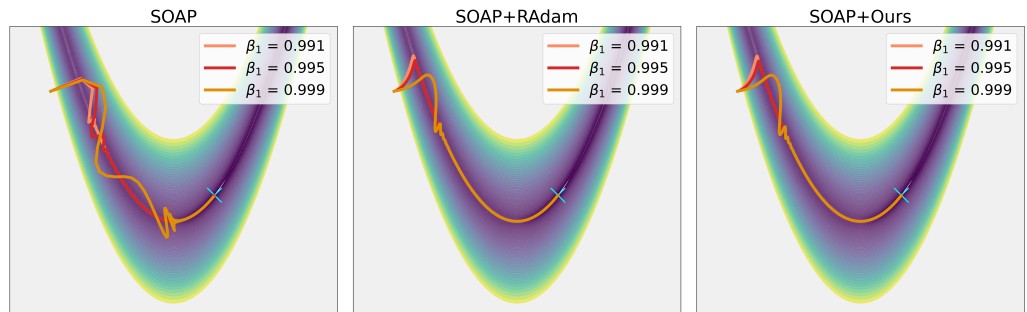

suboptimal across training stages, as early phases require more conservative corrections to prevent divergence; and (3) uniform rectification across parameters ignores significant scale heterogeneity across layers, resulting in suboptimal updates.

To address these limitations, we introduce **Adaptive Confidence Rectification** (ACR), a novel algorithm that adjusts second-moment estimates using an empirical measure of their statistical reliability. Unlike RAdam, which applies a fixed variance correction derived from a specific distributional model, ACR dynamically modulates rectification strength according to the observed variability of the second-moment estimates. This data-driven strategy yields a more robust and flexible optimization process for complex, nonstationary training landscapes. By leveraging an empirical confidence metric (Efron & Tibshirani, 1993), ACR captures both moment stability and parameter-specific scales, enabling principled variance adaptation throughout training.

**Our contributions are threefold:**

1. **Methodological:** We propose *Adaptive Confidence Rectification* (ACR), a framework for dynamically correcting second-moment gradient estimates using an empirical measure of statistical reliability, enabling principled, data-driven variance adaptation (Section 4).

2. **Algorithmic:** We instantiate ACR for several second-order adaptive optimizers (SOAP, AdaHessian, Sophia). The rectified variants incorporate confidence-aware adjustments to stabilize second-order optimization from the outset; SOAP is used as a motivating example due to its strong empirical performance (Algorithm 1).

3. **Empirical:** We evaluate ACR on a comprehensive suite of PDE-driven PINN tasks—Allen–Cahn, Wave, Korteweg–de Vries, Burgers, Poisson, 2D Wave, and Navier–Stokes—across multiple model scales, architectures, and second-order optimizers. In these settings, ACR produces consistent, significant improvements over RAdam and non-rectified baselines. Detailed results appear in Section 5, Figures 3–6, and Tables 1–3.

## 2 RELATED WORKS

Our approach is closely related to adaptive optimization methods, particularly RAdam, due to its variance rectification mechanism. It is also strongly connected to second-order methods, such as SOAP, and their integration into PINNs. Additional discussions are provided in Section F.

**Adaptive momentum-based methods.** Among first-order optimizers, Adam (Kingma & Ba, 2015) remains the de facto standard, utilizing adaptive learning rates and exponential moving averages of gradients. RAdam (Liu et al., 2020) improves upon Adam by introducing a variance rectification term that stabilizes training in early iterations derived by using a gaussian distribution assumptions. In contrast, our method, rectifies the second-moment estimates based on an empirical measure of their statistical reliability.

**Adaptive second-order methods.** SOAP (Vyas et al., 2025) approximates the empirical Fisher information matrix using two Kronecker-factored preconditioners and enhances efficiency by projecting them into the eigenspace. Sophia (Liu et al., 2024a) instead estimates the diagonal of the Gauss-Newton matrix to enable scalable preconditioning. Our method diverges from SOAP by explicitly rescaling the adaptive learning rate to control its variance during the initial training phases.

**Second-order methods for PINNs.** Second-order methods have recently gained traction in the context of PINNs. Müller & Zeinhofer (2023) introduced the Energy Natural Gradient (ENG) method, which is equivalent to a Newton method in function space. Schwencke & Furtlehner (2025) extend this line of work by leveraging the functional geometry induced by the Neural Tangent Kernel (NTK), while Jnini & Vella (2024) and Jnini et al. (2024) propose Hessian-free and Gauss-Newton methods, respectively, tailored for functional optimization in scientific computing tasks. A comprehensive taxonomy is provided by Müller & Zeinhofer (2024). Our method shares the goal of approximating second-order information but departs from prior work in three key aspects: (1) it is compatible with minibatch training, (2) it supports a layerwise preconditioning structure, and (3) it integrates momentum-style exponential moving averages (EMAs) of both first- and second-moment gradient statistics.

## 3 BACKGROUND

We briefly review the PINN training objective (Raissi et al., 2019), the RAdam optimizer (Liu et al., 2020), and the SOAP framework (Vyas et al., 2025). Our proposed method, introduced in the following section, extends the rectification procedure of RAdam and provides a general improvement applicable to adaptive optimizers such as SOAP (Vyas et al., 2025), AdaHessian (Yao et al., 2021), Sophia (Liu et al., 2024a).

**PINN Setup.** We consider a general system of partial differential equations (PDEs) of the form $\mathbf{u}_t + \mathcal{N}[\mathbf{u}] = 0$ for $t \in [0, T]$ and $\mathbf{x} \in \Omega$, subject to initial and boundary conditions: $\mathbf{u}(0, \mathbf{x}) = \mathbf{g}(\mathbf{x})$ for $\mathbf{x} \in \Omega$, and $\mathcal{B}[\mathbf{u}] = 0$ for $t \in [0, T]$, $\mathbf{x} \in \partial\Omega$, where $\mathcal{N}[\cdot]$ denotes a nonlinear differential operator and $\mathcal{B}[\cdot]$ enforces boundary constraints. PINNs approximate the solution $\mathbf{u}(t, \mathbf{x})$ by a neural network $\mathbf{u}_\theta(t, \mathbf{x})$ parameterized by $\theta$. The PDE residual and its corresponding loss over a set of residual points $\{(t_r^i, \mathbf{x}_r^i)\}_{i=1}^{N_r}$ are defined as

$$\mathcal{R}[u_\theta](t, \mathbf{x}) = \frac{\partial u_\theta}{\partial t}(t, \mathbf{x}) + \mathcal{N}[u_\theta](t, \mathbf{x}), \tag{1}$$

$$\mathcal{L}_{\text{PDE}}(\theta) = \frac{1}{N_r} \sum_{i=1}^{N_r} \left| \mathcal{R}[u_\theta](t_r^i, \mathbf{x}_r^i) \right|^2. \tag{2}$$

The overall stochastic gradient for a mini-batch $x \in X$ integrates contributions from all components of the loss:

$$\mathbf{g} = \nabla_\theta \left[ \lambda_{\text{IC}} \, \mathcal{L}_{\text{IC}}(\theta; x_{\text{IC}}) + \lambda_{\text{BC}} \, \mathcal{L}_{\text{BC}}(\theta; x_{\text{BC}}) + \lambda_{\text{PDE}} \, \mathcal{L}_{\text{PDE}}(\theta; x_{\text{PDE}}) \right], \tag{3}$$

where $x_{\text{IC}}$, $x_{\text{BC}}$, and $x_{\text{PDE}}$ are independent mini-batches sampled from the initial condition, boundary condition, and PDE residual datasets, respectively.

**Momentum-based Methods.** The work of Wang et al. (2025), which aligns closely with our objectives, demonstrates that second-order optimization methods—particularly SOAP—effectively mitigate conflicts among competing gradient signals from multiple loss terms. By promoting gradient

alignment, these techniques significantly enhance both the optimization trajectory and the generalization performance of PINNs. We build upon this foundation to further improve the performance of second-order methods for PINNs by robust rectification.

**RAdam.** Liu et al. (2020) identify a critical limitation of the Adam optimizer during the initial training phase: the adaptive learning rate $\boldsymbol{\nu}^{(t)} = \beta_2 \boldsymbol{\nu}^{(t-1)} + (1 - \beta_2)\mathbf{g}^{(t)^2}$ suffers from high variance, which hampers convergence. RAdam introduces a variance rectification mechanism that stabilizes this adaptive term, leading to improved convergence behavior and greater robustness during initialization.

$$
\begin{cases}
\mathbf{m}^{(t)} = \beta_1 \mathbf{m}^{(t-1)} + (1 - \beta_1)\mathbf{g}^{(t)}, \qquad \hat{\mathbf{m}}^{(t)} = \frac{\mathbf{m}^{(t)}}{1 - \beta_1^t} \\
\boldsymbol{\nu}^{(t)} = \beta_2 \boldsymbol{\nu}^{(t-1)} + (1 - \beta_2)\mathbf{g}^{(t)^2}, \qquad \hat{\boldsymbol{\nu}}^{(t)} = \frac{\boldsymbol{\nu}^{(t)}}{1 - \beta_2^t} \\
\rho_t \leftarrow \rho_\infty - 2t\beta_2^t/(1 - \beta_2^t) \\
r_t \leftarrow \sqrt{\frac{(\rho_t - 4)(\rho_t - 2)\rho_\infty}{(\rho_\infty - 4)(\rho_\infty - 2)\rho_t}} \\
\boldsymbol{\theta}^{(t)} = \boldsymbol{\theta}^{(t-1)} - \eta\left(r_t \frac{\hat{\mathbf{m}}^{(t)}}{\sqrt{\hat{\boldsymbol{\nu}}^{(t)}} + \epsilon} + \lambda \boldsymbol{\theta}^{(t-1)}\right).
\end{cases} \tag{RAdam}
$$

**SOAP.** SOAP (Vyas et al., 2025) approximates the full layerwise Hessian by accumulating outer products of gradients (i.e., $\mathbf{g}\mathbf{g}^\top$, $\mathbf{g}^\top\mathbf{g}$), and iteratively refines a preconditioner by updating an orthogonal basis derived from previous estimates. The method projects both the gradients and the exponential moving average (EMA) into the eigenspace, performs updates in this transformed space, and then projects back to the original space. The algorithm proceeds as follows:

$$
\begin{cases}
L \leftarrow \beta_2 L + (1 - \beta_2)\mathbf{g}\mathbf{g}^\top \quad R \leftarrow \beta_2 R + (1 - \beta_2)\mathbf{g}^\top\mathbf{g} \\
Q_L = \text{eigenvectors}(L, Q_L) \quad Q_R = \text{eigenvectors}(R, Q_R) \\
\mathbf{m}^{(t)} = \beta_1 \mathbf{m}^{(t-1)} + (1 - \beta_1)\mathbf{g}^{(t)}, \qquad \hat{\mathbf{m}}^{(t)} = \frac{\mathbf{m}^{(t)}}{1 - \beta_1^t} \\
\mathbf{m}'^{(t)} \leftarrow Q_L^\top \hat{\mathbf{m}}^{(t)} Q_R \\
\mathbf{g}'^{(t)} \leftarrow Q_L^\top \mathbf{g}^{(t)} Q_R \\
\boldsymbol{\nu}^{(t)} = \beta_2 \boldsymbol{\nu}^{(t-1)} + (1 - \beta_2)\mathbf{g}'^{(t)^2}, \qquad \hat{\boldsymbol{\nu}}^{(t)} = \frac{\boldsymbol{\nu}^{(t)}}{1 - \beta_2^t} \\
\boldsymbol{\theta}^{(t)} = \boldsymbol{\theta}^{(t-1)} - \eta\left(Q_L \frac{\mathbf{m}'^{(t)}}{\sqrt{\hat{\boldsymbol{\nu}}^{(t)}} + \epsilon} Q_R^\top + \lambda \boldsymbol{\theta}^{(t-1)}\right)
\end{cases} \tag{SOAP}
$$

# 4 ADAPTIVE CONFIDENCE RECTIFICATION (ACR)

**The issue of Training instability.** Adaptive optimizers often exhibit high variance in learning-rate scaling, particularly in the early stages of training. This instability arises because their second-moment gradient estimates—which determine the adaptive learning rate—are unreliable when computed from a limited number of updates. RAdam (Liu et al., 2020) mitigates this issue by introducing a variance rectification term; however, its correction relies on the assumption that gradients are drawn from a zero-centered normal distribution.

**Limitations of RAdam.** RAdam's rectification is derived from a theoretically motivated effective length of the moving average, $\rho_t$, which depends critically on the presumed gradient distribution. In practice, this approach has three key shortcomings: (1) gradient distributions in deep networks are often heavy-tailed or otherwise non-Gaussian, undermining the validity of the theoretical correction; (2) the fixed variance target used by RAdam may be overly aggressive during early training, when a more conservative adjustment is desirable to prevent divergence; and (3) a single, uniform rectification factor fails to account for the wide variation in parameter scales across layers, leading to suboptimal updates.

**Proposed method.** We introduce *Adaptive Confidence Rectification* (ACR), an alternative that stabilizes second-moment estimates by directly assessing their empirical reliability. Rather than relying on rigid distributional assumptions, ACR uses the observed variability of the second-moment statistics to dynamically modulate the rectification strength. This design yields a more robust, data-driven optimization process capable of adapting to diverse training regimes. Specifically, ACR employs an empirical confidence metric (Efron & Tibshirani, 1993) to quantify the stability of gradient moments, enabling layer-wise rectification that naturally reflects both the observed noise level and the intrinsic scale of each parameter group.

## 4.1 METHODOLOGY

**Confidence Estimation**  To quantify the reliability of our second-moment estimates, we track their empirical variability. We introduce a state variable, $s_t$, which maintains an exponentially decaying average of the squared differences between consecutive second-moment estimates, $v_t$ and $v_{t-1}$:

$$s_t = \beta_s s_{t-1} + (1 - \beta_s)(v_t - v_{t-1})^2 \qquad (4)$$

where $\beta_s = 0.99$ is a decay rate. A smaller value of $s_t$ indicates that the estimate is more stable and, therefore, more reliable. We normalize this variability using the coefficient of variation (Gelman et al., 2014), which provides a scale-invariant measure of uncertainty:

$$CV_t = \frac{\sqrt{s_t}}{v_t + \epsilon} \qquad (5)$$

This allows for a consistent comparison of uncertainty across different parameters. From this, we define a confidence factor, $\text{conf}_t$, which maps the uncertainty to a bounded range, where a value closer to 1 signifies greater statistical stability:

$$\text{conf}_t = \frac{1}{1 + CV_t} \in [0, 1] \qquad (6)$$

**Adaptive Target Selection**  Instead of using RAdam's fixed $\rho_t$, ACR computes an adaptive target that interpolates between the standard RAdam value and a conservative fallback. This interpolation is governed by our confidence factor, $\text{conf}_t$.

$$\rho_{\text{target}} = \text{conf}_t \cdot \rho_t + (1 - \text{conf}_t) \cdot \rho_{\text{safe}} \qquad (7)$$

The standard $\rho_t$ is given by RAdam's formulation, $\rho_t = \rho_\infty - \frac{2t\beta_2^t}{1 - \beta_2^t}$, and $\rho_{\text{safe}} = 10$ serves as a stable, conservative value to fall back on during periods of low confidence. This mechanism ensures that during the unstable early stages of training, ACR's rectification is more conservative, transitioning smoothly to a more aggressive state as the estimates become more reliable.

**Gradient-Aware Scaling**  To account for the significant variance in parameter scales across different network layers, we incorporate a gradient magnitude normalization. This provides a way to modulate the rectification based on the parameter's role in the optimization process. We define a gradient-aware scaling factor, $\gamma_t$, as:

$$\gamma_t = \frac{\|\hat{m}_t\|_2}{\|\hat{m}_t\|_2 + \tau\sqrt{d}} \qquad (8)$$

where $\hat{m}_t$ is the bias-corrected first-moment estimate, $d$ is the parameter dimension and $\tau$ is a tunable scale parameter. This factor allows the rectification to be more pronounced for parameters with large gradient magnitudes, which are often the primary drivers of optimization.

**ACR Rectification Coefficient and Update Rule**  The base rectification coefficient for ACR, $r_{\text{base}}$, follows the structure of RAdam but uses our adaptively selected target, $\rho_{\text{target}}$:

$$r_{\text{base}} = \sqrt{\frac{(\rho_{\text{target}} - 4)(\rho_{\text{target}} - 2)\rho_\infty}{(\rho_\infty - 4)(\rho_\infty - 2)\rho_{\text{target}}}} \qquad (9)$$

The final ACR coefficient, $r_t^{\text{ACR}}$, is a product of this base term and our gradient-aware scaling factor:

$$r_t^{\text{ACR}} = r_{\text{base}} \cdot (1 + \lambda\gamma_t) \qquad (10)$$

where $\lambda$ controls the strength of the scaling. The parameter update rule then applies this coefficient conditionally, preventing updates when the statistical reliability is below a certain threshold:

$$\theta_t = \theta_{t-1} - \alpha_t \cdot \begin{cases} r_t^{\text{ACR}} \cdot \frac{\hat{m}_t}{\sqrt{\hat{v}_t} + \epsilon} & \text{if } \rho_{\text{target}} > 4 \\ \frac{\hat{m}_t}{\sqrt{\hat{v}_t} + \epsilon} & \text{otherwise} \end{cases} \qquad (11)$$

This conditional update ensures stable behavior and robust convergence.

**Properties.** Key properties of ACR: 1) its **adaptive confidence mechanism** balances aggressive optimization with conservative updates, 2) its **scale-awareness** allows effective handling of varying dynamics across different network layers, and 3) its **conservative fallback mechanism** preserves compatibility with RAdam in high-confidence scenarios.

**Hyperparameters.** We did not fine-tune the following standard hyperparameters. $\beta_s = 0.99$ balances responsiveness and stability, similar to $\beta_2$ in Adam, capturing gradient stability without excessive noise. $\tau = 0.1$ acts as a stabilizer in gradient normalization, preventing numerical instability and ensuring scale invariance across parameters. $\rho_{\text{safe}} = 10$ provides a conservative fallback above the critical threshold ($\rho = 4$), ensuring stability during early training.

## 4.2 ALGORITHM

Algorithm 1 illustrates the application of our method to SOAP (Vyas et al., 2025), the best-performing baseline in our experiments. We demonstrate how ACR can be combined with Sophia (Liu et al., 2024a) in Appendix E and with AdaHessian (Yao et al., 2021) in Appendix D. For completeness, we also describe SOAP+RAdam in Appendix C.

---

**Algorithm 1** Our method: Adaptive Confidence Rectification for SOAP. Differences are in blue.

---

**Require:** Data distribution $D$. Initial model parameters $\boldsymbol{\theta}^{(0)}$. Learning rate $\eta$. $\epsilon$ a small constant. SOAP parameters: $\beta_1, \beta_2$ and $\lambda$. $\mathbf{x} \sim D$ data. ACR parameters: $\beta_s 0.99, \tau = 0.1$ and $\rho_{\text{safe}} = 10$.

1: $\mathbf{m}^{(0)} \leftarrow \mathbf{0}, \boldsymbol{\nu}^{(0)} \leftarrow \mathbf{0}$          ▷ Initialize EMAs

2: $s^{(0)} \leftarrow 0$          ▷ Initialize variance tracker for ACR

3: $\mathbf{g}^{(t)} \leftarrow \nabla_{\boldsymbol{\theta}} \mathcal{L}_{\boldsymbol{\theta}^{(t-1)}}(\mathbf{x})$          ▷ Compute gradient

    **Project into Eigenspace**

4: $\mathbf{m}^{(t)} = \beta_1 \mathbf{m}^{(t-1)} + (1 - \beta_1)\mathbf{g}^{(t)}, \qquad \hat{\mathbf{m}}^{(t)} = \frac{\mathbf{m}^{(t)}}{1 - \beta_1^t}$      ▷ Update EMA

5: $\mathbf{m}'^{(t)} \leftarrow Q_L^{\top} \hat{\mathbf{m}}^{(t)} Q_R$          ▷ Rotate EMA

6: $\mathbf{g}'^{(t)} \leftarrow Q_L^{\top} \mathbf{g}^{(t)} Q_R$          ▷ Rotate gradients

    **Adam-style update**

7: $\boldsymbol{\nu}^{(t)} = \beta_2 \boldsymbol{\nu}^{(t-1)} + (1 - \beta_2)\mathbf{g}'^{(t)^2}, \qquad \hat{\boldsymbol{\nu}}^{(t)} = \frac{\boldsymbol{\nu}^{(t)}}{1 - \beta_2^t}$      ▷ Update second moment EMA

8: $\rho_{\infty} = 2/(1 - \beta_2)$

9: $\rho_t = \rho_{\infty} - \frac{2t\beta_2^t}{1 - \beta_2^t}$          ▷ Compute length of approximated SMA

    **ACR: Adaptive Confidence Rectification**

10: $s^{(t)} = \beta_s s^{(t-1)} + (1 - \beta_s)(\hat{\boldsymbol{\nu}}^{(t)} - \hat{\boldsymbol{\nu}}^{(t-1)})^2$          ▷ Update variance tracker

11: $CV_t = \frac{\sqrt{s^{(t)}}}{\hat{\boldsymbol{\nu}}^{(t)} + \epsilon}$          ▷ Coefficient of variation

12: $\text{conf}_t = \frac{1}{1 + CV_t}$          ▷ Compute confidence factor

13: $\rho_{\text{target}} = \text{conf}_t \cdot \rho_t + (1 - \text{conf}_t) \cdot \rho_{\text{safe}}$          ▷ Adaptive target

14: $\gamma_t = \frac{\|\hat{\mathbf{m}}^{(t)}\|_2}{\|\hat{\mathbf{m}}^{(t)}\|_2 + \tau\sqrt{d}}$          ▷ Compute gradient-aware factor

15: $r_{\text{base}} = \sqrt{\frac{(\rho_{\text{target}} - 4)(\rho_{\text{target}} - 2)\rho_{\infty}}{(\rho_{\infty} - 4)(\rho_{\infty} - 2)\rho_{\text{target}}}}$          ▷ Base rectification coefficient

16: $r_t^{\text{ACR}} = r_{\text{base}} \cdot (1 + \lambda\gamma_t)$          ▷ Final ACR coefficient

17: **if** $\rho_{\text{target}} > 4$ **then**

18:      $N' = r_t^{\text{ACR}} \frac{\mathbf{m}'^{(t)}}{\sqrt{\hat{\boldsymbol{\nu}}^{(t)}} + \epsilon}$          ▷ Adapted and rectified momentum

19: **else**

20:      $N' = \frac{\mathbf{m}'^{(t)}}{\sqrt{\hat{\boldsymbol{\nu}}^{(t)}} + \epsilon}$          ▷ un-adapted momentum

21: **end if**

22: $N \leftarrow Q_L N' Q_R^{\top}$          ▷ Rotate the updates back to original space

23: $\boldsymbol{\theta}^{(t)} = \boldsymbol{\theta}^{(t-1)} - \eta(N + \lambda\boldsymbol{\theta}^{(t-1)})$          ▷ Update weights

    **Update preconditioners**

24: $L \leftarrow \beta_2 L + (1 - \beta_2)\mathbf{g}\mathbf{g}^{\top}, \quad R \leftarrow \beta_2 R + (1 - \beta_2)\mathbf{g}^{\top}\mathbf{g}$      ▷ Approximate hessians

25: **if** $t\%f == 0$ **then**

26:      $Q_L \leftarrow \text{eigenvec}(L, Q_L), \quad Q_R \leftarrow \text{eigenvec}(R, Q_R)$      ▷ Approximate eigenbasis

27: **end if**

---

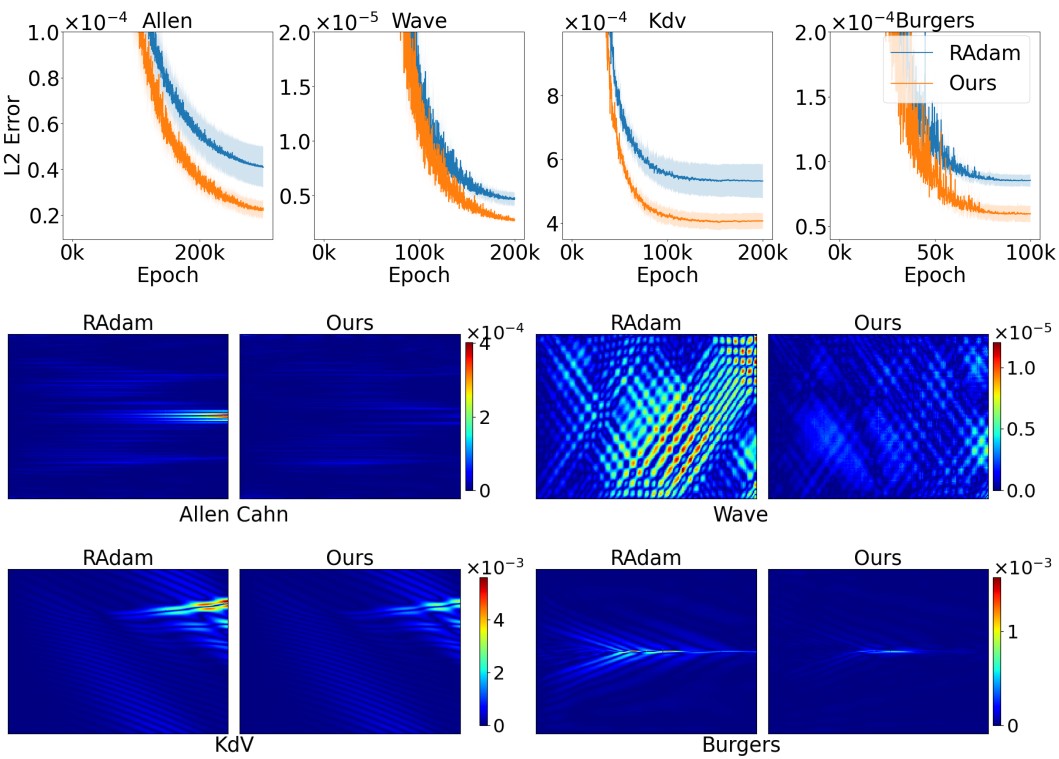

Figure 3: Top: L2 over all datasets. Bottom: Absolute error analysis.

## 5 EXPERIMENTS

We conduct a comprehensive set of experiments to evaluate the effectiveness of our rectification approach in improving state-of-the-art second-order adaptive methods for solving PDEs with PINNs. Our primary baseline is RAdam (Liu et al., 2020). Unless otherwise noted, we apply the rectification to SOAP (Vyas et al., 2025), which currently achieves state-of-the-art performance for PINNs (Wang et al., 2025). We further benchmark against AdaHessian (Yao et al., 2021) and Sophia (Liu et al., 2024a), and assess robustness across multiple PINN architectures, including MLP, ResNet, and PirateNet (Wang et al., 2025) (see Appendix B.2).

**Experimental setup.** To ensure statistical robustness, all results are averaged over five independent runs. Experiments are performed with the JaxPi library (Wang et al., 2024) across four canonical PDEs: Allen–Cahn, Wave, KdV, and Burgers (dataset details provided in Appendix B.1). Hyperparameters are fixed at $\beta_1 = 0.99$ and $\beta_2 = 0.993$, which we identify as consistently optimal across all datasets and model sizes (see Appendix B.3 for additional hyperparameter specifications).

**L2 error across PDE benchmarks.** Figure 3 (top) compares the $L^2$ error progression of our approach versus RAdam over the SOAP algorithm across the four PDE datasets. Our method consistently achieves superior accuracy and faster convergence. Notably, for the Allen–Cahn equation, the $L^2$ error attained by RAdam at epoch 300,000 is matched by our method around epoch 200,000. This trend holds across all PDEs. Final $L^2$ error values at convergence are summarized in Table 1, where our approach outperforms RAdam across the board, without any model-specific fine-tuning—highlighting the robustness of our algorithmic modifications.

**Absolute Error Analysis of Predicted Solutions.** We evaluate the pointwise absolute error between the predicted and ground-truth fields (Wang et al., 2025; Müller & Zeinhofer, 2023). Figure 3 (bottom) presents absolute error maps computed using the final model parameters for each of the four PDE benchmarks. Our method SOAP+ACR consistently shows darker maps against SOAP+RAdam, indicating lower errors.

Table 1: L2 results over multiple model sizes with MLP architecture.

| Dataset | 70K parameters | | 135K parameters | | 200K parameters | |
|---|---|---|---|---|---|---|
| | RAdam | Ours | RAdam | Ours | RAdam | Ours |
| Allen Cahn ($\times10^{-6}$) | 41.0±8.6 | 22.1±3.7 | 5.46±0.99 | 4.79±0.65 | 5.27±1.32 | 3.48±0.05 |
| Wave ($\times10^{-6}$) | 4.70±0.60 | 2.71±0.21 | 3.40±0.26 | 2.65±0.13 | 3.66±0.21 | 2.56±0.24 |
| KdV ($\times10^{-4}$) | 5.32±0.53 | 4.08±0.25 | 4.29±0.18 | 3.28±0.01 | 3.48±0.16 | 3.38±0.26 |
| Burgers ($\times10^{-5}$) | 8.55±0.49 | 5.96±0.66 | 6.60±0.47 | 4.55±0.24 | 4.90±0.03 | 4.11±0.02 |

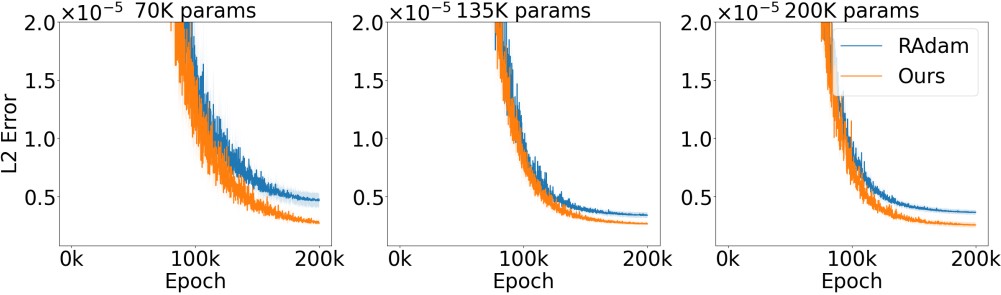

Figure 4: Multiple model sizes over Wave PDE dataset.

**Scalability across model sizes.** To evaluate robustness with respect to model capacity, we test three MLP configurations over the SOAP algorithm: *small* (70K parameters), *medium* (135K parameters), and *large* (200K parameters), corresponding to architectures with one, three, and four hidden layers, respectively. Figure 4 presents the training trajectories on the Wave and KdV equations over 200,000 epochs. Our method consistently delivers lower $L^2$ errors compared to RAdam rectification, regardless of model size. Full results across all datasets and scales are reported in Table 1.

**Robustness across PINN architectures.** We consider PINN architure methods, including MLP (Wang et al., 2025), PirateNet (Wang et al., 2024), ResNet (Wang et al., 2024). We demonstrate for the SOAP + rectification (Algorithm 1), over such architectures. Note that PirateNet and ResNet are significant larger compared to MLP. In Table 2 we show the L2 error results. In Figure 5 we show the full training loss for the KdV PDE. Our method consistently outperforms RAdam rectification method over all cases.

Table 2: L2 results over multiple PINN architectures.

| Dataset | MLP | | ResNet | | PirateNet | |
|---|---|---|---|---|---|---|
| | RAdam | Ours | RAdam | Ours | RAdam | Ours |
| Allen Cahn ($\times10^{-6}$) | 41.0±8.6 | 22.1±3.7 | 6.25±0.61 | 4.40±0.05 | 4.59±0.18 | 4.02±0.05 |
| Wave ($\times10^{-6}$) | 4.70±0.60 | 2.71±0.21 | 3.58±0.54 | 2.52±0.02 | 2.32±0.89 | 1.73±0.13 |
| KdV ($\times10^{-4}$) | 5.32±0.53 | 4.08±0.25 | 3.09±0.08 | 2.68±0.19 | 3.74±0.07 | 3.10±0.18 |
| Burgers ($\times10^{-5}$) | 8.55±0.49 | 5.96±0.66 | 7.94±1.19 | 2.96±0.18 | 6.55±0.02 | 5.14±0.10 |

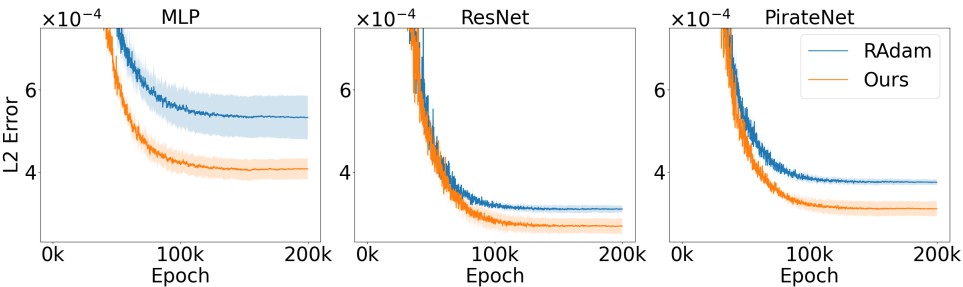

Figure 5: Multiple PINN architectures over KdV PDE dataset.

Table 3: L2 results over multiple optimization algorithms.

| Dataset | SOAP | | AdaHessian ($\times 10^1$) | | Sophia ($\times 10^2$) | |
|---|---|---|---|---|---|---|
| | RAdam | Ours | RAdam | Ours | RAdam | Ours |
| Allen Cahn ($\times 10^{-6}$) | 41.0±8.6 | 22.1±3.7 | 7.84±0.78 | 6.58±0.54 | 4.11±0.50 | 2.64±0.25 |
| Wave ($\times 10^{-6}$) | 4.70±0.60 | 2.71±0.21 | 4.64±0.27 | 3.16±0.07 | 8.19±0.82 | 3.87±1.36 |
| KdV ($\times 10^{-4}$) | 5.32±0.53 | 4.08±0.25 | 9.19±1.26 | 1.78±0.24 | 7.44±0.14 | 6.63±0.06 |
| Burgers ($\times 10^{-5}$) | 8.55±0.49 | 5.96±0.66 | 3.85±1.51 | 1.25±0.14 | 9.00±0.40 | 7.56±0.62 |

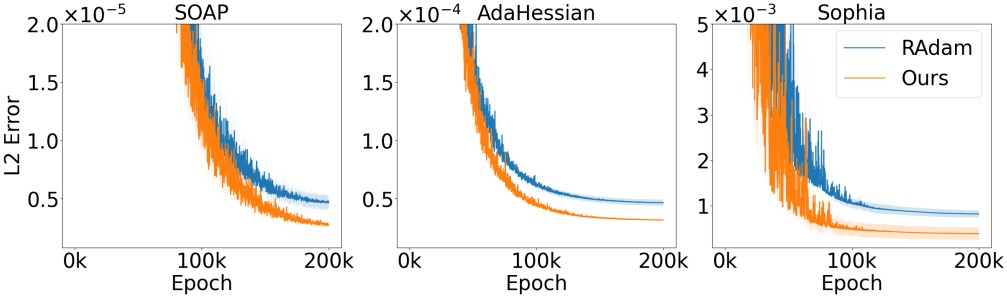

Figure 6: Multiple optimization methods over Wave PDE dataset.

**Robustness across optimization algorithms.** We evaluate the proposed rectification procedure across a diverse set of second-order adaptive optimizers, including SOAP (Vyas et al., 2025) (Algorithm 1), AdaHessian (Yao et al., 2021) (Algorithm 3), and Sophia (Liu et al., 2024a) (Algorithm 4). Our method integrates seamlessly with each optimizer by appropriately rescaling the ratio of moments. As reported in Table 3, it consistently achieves lower L2 errors than the RAdam rectification across all settings, and Figure 6 shows faster convergence, demonstrating strong robustness and broad applicability.

**Comparison with non-rectified variants.** Although our primary objective is to improve upon the RAdam rectification, we further examine whether the proposed strategy also outperfoms non-rectified baselines. We conduct two complementary experiments. Section A.1 reports results on SOAP across multiple model sizes, while Section A.2 extends the comparison to multiple optimizers (AdaHessian, Sophia, and SOAP). In both scenarios, our approach yields substantial performance gains, underscoring the effectiveness and generality of the proposed rectification mechanism.

## 6    CONCLUSION

This work explored the rectification of adaptive learning rates as a means to stabilize the training of physics-informed neural networks (PINNs) and introduced a principled enhancement to the widely used RAdam strategy.

We proposed Adaptive Confidence Rectification (ACR), a novel optimization framework that strengthens variance rectification through a confidence-driven mechanism. In contrast to RAdam, which relies on restrictive distributional assumptions, ACR dynamically adjusts the rectification strength based on an empirical measure of statistical reliability. This formulation enables robust adaptation to heterogeneous gradient landscapes and varying parameter scales, offering both theoretical rigor and practical flexibility.

Extensive experiments on PINNs for solving partial differential equations (PDEs) demonstrate that ACR consistently improves convergence stability and optimization efficiency over RAdam and non-rectified variants of SOAP, Sophia, and AdaHessian. These gains persist across diverse model sizes, PINN architectures, and second-order adaptive optimizers, underscoring the broad applicability and effectiveness of the proposed method.

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

# A ADDITIONAL EXPERIMENTAL RESULTS

## A.1 COMPARISON WITH NON-RECTIFIED OVER MULTIPLE MODEL SIZES

In Table 4 we report the L2 error results for our SOAP without rectification and with our method.

Table 4: L2 results. Our method achieves lower error over all considered datasets and model sizes.

| Dataset | 70K parameters | | 135K parameters | | 200K parameters | |
|---|---|---|---|---|---|---|
| | SOAP | Ours | SOAP | Ours | SOAP | Ours |
| Allen Cahn ($\times 10^{-6}$) | 42.3±4.00 | 22.1±3.7 | 5.73±0.26 | 4.79±0.65 | 4.93±0.02 | 3.48±0.05 |
| Wave ($\times 10^{-6}$) | 4.98±0.56 | 2.71±0.21 | 2.85±0.24 | 2.65±0.13 | 3.60±0.13 | 2.56±0.24 |
| KdV ($\times 10^{-4}$) | 5.71±0.45 | 4.08±0.25 | 4.44±0.22 | 3.28±0.01 | 4.20±0.50 | 3.38±0.26 |
| Burgers ($\times 10^{-5}$) | 8.42±0.18 | 5.96±0.66 | 7.28±0.78 | 4.55±0.24 | 4.60±0.01 | 4.11±0.02 |

## A.2 COMPARISON WITH NON-RECTIFIED OVER MULTIPLE MODEL SIZES

We compare our rectification procedure against the baseline withut rectification, over multiple second-order adaptive methods, including SOAP (Vyas et al., 2025) (Algorithm 1), AdaHessian (Yao et al., 2021) (Algorithm 3), Sophia (Liu et al., 2024a) (Algorithm 4). In Table 5 we show the L2 error results. The ACR method consistently outperfoms the RAdam rectification over the considered cases.

Table 5: L2 results over multiple optimization algorithms.

| Dataset | SOAP | | AdaHessian ($\times 10^1$) | | Sophia($\times 10^2$) | |
|---|---|---|---|---|---|---|
| | Non-rectif | Ours | Non-rectif | Ours | Non-rectif | Ours |
| Allen Cahn ($\times 10^{-6}$) | 42.3±4.00 | 22.1±3.7 | 8.6±0.58 | 6.58±0.54 | 3.83±0.81 | 2.64±0.25 |
| Wave ($\times 10^{-6}$) | 4.98±0.56 | 2.71±0.21 | 4.4±0.21 | 3.16±0.07 | 7.58±1.19 | 3.87±1.36 |
| KdV ($\times 10^{-4}$) | 5.71±0.45 | 4.08±0.25 | 3.61±0.40 | 1.78±0.24 | 8.78±1.17 | 6.63±0.06 |
| Burgers ($\times 10^{-5}$) | 8.42±0.18 | 5.96±0.66 | 2.89±0.86 | 1.25±0.14 | 10.8±0.10 | 7.56±0.62 |

## B EXPERIMENTAL SETTINGS

The experiments from Section 5 follow the setup from (Wang et al., 2025). Below we describe data generation and the governing PDEs, architecture, hyperparameters.

### B.1 DATASETS

**Data Generation.** The reference datasets were generated using the numerical package Chebfun in MATLAB. The data was initially simulated with a fine time resolution of $dt = 10^{-4}$, and then temporally downsampled to produce the final dataset used for analysis. Table 6 summarizes the parameters of the partial differential equations (PDEs) and key details of the generated datasets.

Table 6: Parameter and numerical configurations for generating the PDE solutions.

| PDE | Parameter | Package | Resolution |
|---|---|---|---|
| **Allen Cahn** | $\epsilon = 10^{-4}, a = 5$ | Chebfun | $200 \times 512$ |
| **Wave** | $c = 4$ | N/A | $200 \times 128$ |
| **KdV** | $\eta = 1, \mu = 0.022$ | Chebfun | $200 \times 512$ |
| **Burgers** | $\nu = 0.01\,\pi$ | Chebfun | $200 \times 512$ |

**Allen-Cahn equation.** We investigate the one-dimensional Allen-Cahn equation with periodic boundary conditions:

$$u_t - 0.0001u_{xx} + 5u^3 - 5u = 0, \quad t \in [0, 1], \ x \in [-1, 1],$$

$$u(0, x) = x^2 \cos(\pi x),$$

$$u(t, -1) = u(t, 1), \quad u_x(t, -1) = u_x(t, 1).$$

where $u$ represents the order parameter (e.g., concentration difference between two phases), $\epsilon$ controls the interfacial width, $a$ is the reaction rate coefficient, and the term $(u - u^3)$ drives the phase separation.

**Wave equation.** We consider a one-dimensional wave equation in the domain $\Omega = [0, 1] \times [0, 1]$ taking the form

$$u_{tt}(x, t) - 4u_{xx}(x, t) = 0, \quad (x, t) \in (0, 1) \times (0, 1),$$

$$u(0, t) = u(1, t) = 0, \quad t \in [0, 1],$$

$$u(x, 0) = \sin(\pi x) + \frac{1}{2}\sin(4\pi x), \quad x \in [0, 1],$$

$$u_t(x, 0) = 0, \quad x \in [0, 1].$$

where $u$ represents the wave amplitude, and $c$ is the wave propagation speed, determined by the medium's physical properties.

By d'Alembert's formula, the solution $u(x, t)$ is given by

$$u(x, t) = \sin(\pi x) \cos(2\pi t) + \frac{1}{2}\sin(4\pi x)\cos(8\pi t).$$

**Korteweg-De Vries equation.** The one-dimensional KdV equation is expressed as follows:

$$u_t + \eta u u_x + \mu^2 u_{xxx} = 0, \quad t \in (0, 1), \quad x \in (-1, 1),$$

$$u(x, 0) = \cos(\pi x),$$

$$u(t, -1) = u(t, 1),$$

where $u$ represents the wave amplitude or water surface elevation, and $\eta$ governs the strength of the nonlinearity, while $\mu$ controls the dispersion level. Under the KdV dynamics, this initial wave evolves into a series of solitary-type waves.

**Burgers equation.** The 1D Burgers equation is defined as:

$$u_t + u u_x = \nu u_{xx},$$

where $u$ represents the velocity field, and $\nu$ is the kinematic viscosity coefficient controlling the diffusion strength. Here we set $(x, t) \in \Omega = [-1, 1] \times [0, 1]$, with initial and boundary conditions:

$$u(x, 0) = -\sin(\pi x),$$
$$u(-1, t) = u(1, t) = 0,$$

and viscosity parameter $\nu = 0.01/\pi$.

## B.2 MODEL ARCHITECTURES

In our experiments, we use MLP, PirateNet, Resnet architectures from (Wang et al., 2025).

**MLP.** When comparing multiple datasets, i.e., in Figure 3 (top), we use the MLP with one layer. When comparing over multiple model sizes, i.e., in Figure 3 (bottom) and in Table 1, we compare the MLP with 1,3 and 4 layers.

The used architecture is defined as follows. We use a multilayer perceptron (MLP) defined as a composition of $L$ fully connected layers with hidden dimension $d_h$, where $L = 4$ and $d_h = 256$ by default. The input $x \in \mathbb{R}^n$ is optionally encoded using a positional embedding function $E(\cdot)$, which may include periodic features or Fourier embeddings, resulting in an encoded input $\tilde{x} = E(x)$. Each hidden layer applies a linear transformation followed by a non-linear activation function $\sigma(\cdot)$, typically $\tanh$:

$$h^{(0)} = \tilde{x}, \quad h^{(l)} = \sigma(W^{(l)} h^{(l-1)} + b^{(l)}), \quad \text{for } l = 1, \ldots, L,$$

where $W^{(l)} \in \mathbb{R}^{d_h \times d_h}$ and $b^{(l)} \in \mathbb{R}^{d_h}$ are the weights and biases of layer $l$, possibly reparameterized as specified. The final output $y$ is computed through a final learned linear layer:

$$y = W^{(L+1)} h^{(L)} + b^{(L+1)}.$$

**PirateNet.** The PirateNet is designed to address the initialization problem in neural networks. It features a specialized forward pass with residual blocks and gate-like functions derived from two encoding maps, $U$ and $V$, based on an embedding function $\Phi(x)$. The forward pass for each residual block $l$ is defined as follows:

$$f^{(l)} = \sigma(W_1^{(l)} x^{(l)} + b_1^{(l)}), \tag{12}$$

$$z_1^{(l)} = f^{(l)} \odot U + (1 - f^{(l)}) \odot V, \tag{13}$$

$$g^{(l)} = \sigma(W_2^{(l)} z_1^{(l)} + b_2^{(l)}), \tag{14}$$

$$z_2^{(l)} = g^{(l)} \odot U + (1 - g^{(l)}) \odot V, \tag{15}$$

$$h^{(l)} = \sigma(W_3^{(l)} z_2^{(l)} + b_3^{(l)}), \tag{16}$$

$$x^{(l+1)} = \alpha^{(l)} h^{(l)} + (1 - \alpha^{(l)}) x^{(l)}, \tag{17}$$

where $\odot$ denotes point-wise multiplication, and $\alpha^{(l)}$ is a trainable parameter.

The input coordinates $x$ are first mapped to a high-dimensional feature space using a random Fourier embedding function $\Phi(x) = \begin{bmatrix} \cos(Bx) \\ \sin(Bx) \end{bmatrix}$, with $B$ sampled from a Gaussian distribution. The encoded embeddings $\Phi(x)$ are then sent into two dense layers to create the gate functions $U = \sigma(W_1 \Phi(x) + b_1)$ and $V = \sigma(W_2 \Phi(x) + b_2)$.

The final output of a PirateNet of $L$ residual blocks is given by $\mathbf{u}_\theta = W^{(L+1)} \mathbf{x}^{(L)}$.

**ResNet.** The ResNet for PINNs is an architecture that incorporates residual connections to help mitigate the vanishing gradient problem in deep neural networks. This structure allows for the training of deeper models, which can better approximate complex solutions to partial differential equations (PDEs). The core of the architecture lies in its residual blocks, which add the input of the block to its output.

The forward pass for a single residual block is given by:

$$\mathbf{y}^{(l+1)} = \mathbf{y}^{(l)} + \sigma(W_2^{(l)}(\sigma(W_1^{(l)} \mathbf{y}^{(l)} + \mathbf{b}_1^{(l)})) + \mathbf{b}_2^{(l)}), \tag{18}$$

where $\mathbf{y}^{(l)}$ is the input to the block, $\sigma$ is an activation function, $W_1^{(l)}$ and $W_2^{(l)}$ are the weight matrices, and $\mathbf{b}_1^{(l)}$ and $\mathbf{b}_2^{(l)}$ are the bias vectors.

For a PINN, the input is typically the coordinates $(x, t)$. This input is passed through a series of residual blocks. The input to the first layer, $\mathbf{y}^{(0)}$, is the coordinate vector, i.e., $\mathbf{y}^{(0)} = [x, t]^T$. The output of the final layer, $\mathbf{y}^{(L)}$, is then mapped to the final solution $\mathbf{u}_\theta$ through a linear transformation:

$$\mathbf{u}_\theta(x, t) = W^{(L+1)}\mathbf{y}^{(L)} + \mathbf{b}^{(L+1)}. \tag{19}$$

This output $\mathbf{u}_\theta$ is then used to construct the physics-informed loss function, which consists of a data-driven part and a physics-informed part. The loss function is minimized to train the network.

### B.3 Hyperparameters details

We fine-tune the momentum hyperparameters for SOAP $\beta_1$ and $\beta_2$, then use the same values for our method. We search for $\beta_1$ between 0.96, and 0.99 and $\beta_2$ between 0.990, and 0.999. For learning rate we search between 0.0001 and 0.0015, but we find that $LR = 0.001$ works efficiently across all experiments, so we do not further fine-tune LR for our method. Table 7 reports the parameters used in experiments.

Table 7: Learning hyperparameters.

|  | Soap | | | Ours | | |
|---|---|---|---|---|---|---|
|  | LR | $\beta_1$ | $\beta_2$ | LR | $\beta_1$ | $\beta_2$ |
| Allen Cahn | 0.001 | 0.99 | 0.993 | 0.001 | 0.99 | 0.993 |
| Burgers | 0.001 | 0.99 | 0.993 | 0.001 | 0.99 | 0.993 |
| KDV | 0.001 | 0.99 | 0.993 | 0.001 | 0.99 | 0.993 |
| Wave | 0.001 | 0.99 | 0.993 | 0.001 | 0.99 | 0.993 |

## C  RECTIFIED SOAP WITH RADAM

We provide a rectified version of the baseline SOAP optimizer by integrating the variance rectification mechanism from RAdam (Liu et al., 2020). The resulting algorithm, SOAP + RAdam, preserves the eigenspace preconditioning of SOAP while correcting for the variance of the adaptive learning rate during the early stages of training. The full procedure is summarized in Algorithm 2.

---

**Algorithm 2** SOAP + RAdam Algorithm. Differences with SOAP are in blue.

---

**Require:** Data distribution $D$. Initial model parameters $\boldsymbol{\theta}^{(0)}$. Learning rate $\eta$. $\epsilon$ a small constant. SOAP parameters: $\beta_1$, $\beta_2$ and $\lambda$. $\mathbf{x} \sim D$ data.

1:  $\mathbf{m}^{(0)} \leftarrow \mathbf{0}$ , $\boldsymbol{\nu}^{(0)} \leftarrow \mathbf{0}$                    $\triangleright$ Initialize EMAs

2:  $\mathbf{g}^{(t)} \leftarrow \nabla_{\boldsymbol{\theta}} \mathcal{L}_{\boldsymbol{\theta}^{(t-1)}}(\mathbf{x})$                  $\triangleright$ Compute gradient

3:  $\rho_\infty \leftarrow 2/(1-\beta_2) - 1$        $\triangleright$ Compute max length of the approximated SMA

  **Project into Eigenspace**

4:  $\mathbf{m}^{(t)} = \beta_1 \mathbf{m}^{(t-1)} + (1-\beta_1)\mathbf{g}^{(t)}, \qquad \hat{\mathbf{m}}^{(t)} = \frac{\mathbf{m}^{(t)}}{1-\beta_1^t}$    $\triangleright$ Update EMA

5:  $\mathbf{m}'^{(t)} \leftarrow Q_L^\top \hat{\mathbf{m}}^{(t)} Q_R$                  $\triangleright$ Rotate EMA

6:  $\mathbf{g}'^{(t)} \leftarrow Q_L^\top \mathbf{g}^{(t)} Q_R$                  $\triangleright$ Rotate gradients

  **Adam-style update**

7:  $\boldsymbol{\nu}^{(t)} = \beta_2 \boldsymbol{\nu}^{(t-1)} + (1-\beta_2)\mathbf{g}'^{(t)^2}, \qquad \hat{\boldsymbol{\nu}}^{(t)} = \frac{\boldsymbol{\nu}^{(t)}}{1-\beta_2^t}$  $\triangleright$ Update second moment EMA

8:  $\rho_t \leftarrow \rho_\infty - 2t\beta_2^t/(1-\beta_2^t)$       $\triangleright$ Compute length of approximated SMA

9:  **if** variance is tractable (i.e. $\rho_t > 4$) **then**

10:  $r_t \leftarrow \sqrt{\frac{(\rho_t-4)(\rho_t-2)\rho_\infty}{(\rho_\infty-4)(\rho_\infty-2)\rho_t}}$        $\triangleright$ Compute variance rectification

11:  $N' = r^{(t)} \frac{\mathbf{m}'^{(t)}}{\sqrt{\hat{\boldsymbol{\nu}}^{(t)}}+\epsilon}$         $\triangleright$ Adapted and rectified momentum

12: **else**

13:  $N' = \mathbf{m}'^{(t)}$              $\triangleright$ un-adapted momentum

14: **end if**

15: $N \leftarrow Q_L N' Q_R^\top$         $\triangleright$ Rotate the updates back to original space

16: $\boldsymbol{\theta}^{(t)} = \boldsymbol{\theta}^{(t-1)} - \eta(N + \lambda \boldsymbol{\theta}^{(t-1)})$         $\triangleright$ Update weights

  **Update preconditioners**

17: $L \leftarrow \beta_2 L + (1-\beta_2)\mathbf{g}\mathbf{g}^\top, \quad R \leftarrow \beta_2 R + (1-\beta_2)\mathbf{g}^\top\mathbf{g}$   $\triangleright$ Approximate Hessians

18: **if** $t\%f == 0$ **then**

19:  $Q_L \leftarrow \text{eigenvec}(L, Q_L), \quad Q_R \leftarrow \text{eigenvec}(R, Q_R)$   $\triangleright$ Approximate eigebasis

20: **end if**

---

# D  RECTIFIED ADAHESSIAN WITH ACR

In Algorithm 3, we provide the pseudocode for applying our Adaptive Confidence Rectification (ACR) method to AdaHessian (Yao et al., 2021). This version of AdaHessian uses the Hutchinson estimator to approximate the Hessian diagonal, which is then used as the second-moment estimate. Our ACR method rectifies the update based on the statistical confidence of this diagonal approximation. The ACR framework is general and can be applied to a variety of optimizers, and we found it to be a powerful complement to second-order methods like AdaHessian.

---

**Algorithm 3** Our method: Adaptive Confidence Rectification for AdaHessian. Differences are in blue.

---

**Require:** Data distribution $D$. Initial model parameters $\boldsymbol{\theta}^{(0)}$. Learning rate $\eta$. $\epsilon$ a small constant. AdaHessian parameters: $\beta_1$, $\beta_2$, and $k$. ACR parameters: $\beta_s$, $\tau$ and $\rho_{\text{safe}}$.

1: $\mathbf{m}^{(0)} \leftarrow \mathbf{0}$, $\boldsymbol{\nu}^{(0)} \leftarrow \mathbf{0}$         ▷ Initialize EMAs for moments

2: $s^{(0)} \leftarrow \mathbf{0}$         ▷ Initialize variance tracker for ACR

3: $\mathbf{g}^{(t)} \leftarrow \nabla_{\boldsymbol{\theta}} \mathcal{L}_{\boldsymbol{\theta}^{(t-1)}}(\mathbf{x})$         ▷ Compute gradient

4: $\text{diag}(\mathbf{h}^{(t)}) \leftarrow \text{Hutchinson}(\nabla_{\boldsymbol{\theta}^2} \mathcal{L}(\mathbf{x}))$         ▷ Approximate Hessian diagonal

5: $\mathbf{m}^{(t)} = \beta_1 \mathbf{m}^{(t-1)} + (1 - \beta_1)\mathbf{g}^{(t)}, \qquad \hat{\mathbf{m}}^{(t)} = \frac{\mathbf{m}^{(t)}}{1-\beta_1^t}$         ▷ Update first moment EMA

6: $\boldsymbol{\nu}^{(t)} = \beta_2 \boldsymbol{\nu}^{(t-1)} + (1 - \beta_2)\text{diag}(\mathbf{h}^{(t)}), \qquad \hat{\boldsymbol{\nu}}^{(t)} = \frac{\boldsymbol{\nu}^{(t)}}{1-\beta_2^t}$         ▷ Update second moment EMA

7: $\rho_\infty = 2/(1 - \beta_2)$

8: $\rho_t = \rho_\infty - \frac{2t\beta_2^t}{1-\beta_2^t}$         ▷ Compute length of approximated SMA

**ACR: Adaptive Confidence Rectification**

9: $s^{(t)} = \beta_s s^{(t-1)} + (1 - \beta_s)(\hat{\boldsymbol{\nu}}^{(t)} - \hat{\boldsymbol{\nu}}^{(t-1)})^2$         ▷ Update variance tracker

10: $CV_t = \frac{\sqrt{s^{(t)}}}{\hat{\boldsymbol{\nu}}^{(t)}+\epsilon}$         ▷ Coefficient of variation

11: $\text{conf}_t = \frac{1}{1+CV_t}$         ▷ Compute confidence factor

12: $\rho_{\text{target}} = \text{conf}_t \cdot \rho_t + (1 - \text{conf}_t) \cdot \rho_{\text{safe}}$         ▷ Adaptive target

13: $\gamma_t = \frac{\|\hat{\mathbf{m}}^{(t)}\|_2}{\|\hat{\mathbf{m}}^{(t)}\|_2 + \tau\sqrt{d}}$         ▷ Compute gradient-aware factor

14: $r_{\text{base}} = \sqrt{\frac{(\rho_{\text{target}}-4)(\rho_{\text{target}}-2)\rho_\infty}{(\rho_\infty-4)(\rho_\infty-2)\rho_{\text{target}}}}$         ▷ Base rectification coefficient

15: $r_t^{\text{ACR}} = r_{\text{base}} \cdot (1 + \lambda\gamma_t)$         ▷ Final ACR coefficient

16: **if** $\rho_{\text{target}} > 4$ **then**

17:      $\boldsymbol{\theta}^{(t)} = \boldsymbol{\theta}^{(t-1)} - \eta \cdot r_t^{\text{ACR}} \cdot \frac{\hat{\mathbf{m}}^{(t)}}{\sqrt{\hat{\boldsymbol{\nu}}^{(t)}}^k + \epsilon}$         ▷ Rectified update with Hessian diagonal

18: **else**

19:      $\boldsymbol{\theta}^{(t)} = \boldsymbol{\theta}^{(t-1)} - \eta \cdot \frac{\hat{\mathbf{m}}^{(t)}}{\sqrt{\hat{\boldsymbol{\nu}}^{(t)}}^k + \epsilon}$         ▷ Un-rectified update

20: **end if**

---

# E   RECTIFIED SOPHIA WITH ACR

In Algorithm 4, we provide the pseudocode for applying our Adaptive Confidence Rectification (ACR) method to Sophia (Liu et al., 2024a). This version uses the Hutchinson estimator to approximate the Hessian diagonal, which is then used as the second-moment estimate. Our ACR method rectifies the update based on the statistical confidence of this diagonal approximation. The ACR framework is general and can be applied to a variety of optimizers, and we found it to be a powerful complement to second-order methods like Sophia.

---

**Algorithm 4** Our method: Adaptive Confidence Rectification for Sophia with Clipped Update

---

**Require:** Data distribution $\mathcal{D}$. Initial model parameters $\boldsymbol{\theta}^{(0)}$. Learning rate $\eta$. $\epsilon$ a small constant. Sophia parameters: $\beta_1$, $\beta_2$, and $\gamma$. ACR parameters: $\beta_s$, $\tau$ and $\rho_{\text{safe}}$.

1: $\mathbf{m}^{(0)} \leftarrow \mathbf{0}$ , $\boldsymbol{\nu}^{(0)} \leftarrow \mathbf{0}$            $\triangleright$ Initialize EMAs for moments

2: $s^{(0)} \leftarrow \mathbf{0}$            $\triangleright$ Initialize variance tracker for ACR

3: $\mathbf{g}^{(t)} \leftarrow \nabla_{\boldsymbol{\theta}} \mathcal{L}_{\boldsymbol{\theta}^{(t-1)}}(\mathbf{x})$           $\triangleright$ Compute gradient

4: $\text{diag}(\mathbf{h}^{(t)}) \leftarrow \text{Hutchinson}(\nabla_{\boldsymbol{\theta}}^2 \mathcal{L}(\mathbf{x}))$      $\triangleright$ Approximate Hessian diagonal

5: $\mathbf{m}^{(t)} = \beta_1 \mathbf{m}^{(t-1)} + (1 - \beta_1) \mathbf{g}^{(t)}, \qquad \hat{\mathbf{m}}^{(t)} = \frac{\mathbf{m}^{(t)}}{1 - \beta_1^t}$    $\triangleright$ Update first moment EMA

6: $\boldsymbol{\nu}^{(t)} = \beta_2 \boldsymbol{\nu}^{(t-1)} + (1 - \beta_2)\text{diag}(\mathbf{h}^{(t)}), \qquad \hat{\boldsymbol{\nu}}^{(t)} = \frac{\boldsymbol{\nu}^{(t)}}{1 - \beta_2^t}$   $\triangleright$ Update second moment EMA

7: $\rho_\infty = \frac{2}{1 - \beta_2} - 1$

8: $\rho_t = \rho_\infty - \frac{2t\beta_2^t}{1 - \beta_2^t}$         $\triangleright$ Compute length of approximated SMA

  **ACR: Adaptive Confidence Rectification**

9: $s^{(t)} = \beta_s s^{(t-1)} + (1 - \beta_s)(\hat{\boldsymbol{\nu}}^{(t)} - \hat{\boldsymbol{\nu}}^{(t-1)})^2$      $\triangleright$ Update variance tracker

10: $CV_t = \frac{\sqrt{s^{(t)}}}{\hat{\boldsymbol{\nu}}^{(t)} + \epsilon}$           $\triangleright$ Coefficient of variation

11: $\text{conf}_t = \frac{1}{1 + CV_t}$          $\triangleright$ Compute confidence factor

12: $\rho_{\text{target}} = \text{conf}_t \cdot \rho_t + (1 - \text{conf}_t) \cdot \rho_{\text{safe}}$       $\triangleright$ Adaptive target

13: $\gamma_t = \frac{\|\hat{\mathbf{m}}^{(t)}\|_2}{\|\hat{\mathbf{m}}^{(t)}\|_2 + \tau\sqrt{d}}$        $\triangleright$ Compute gradient-aware factor

14: $r_{\text{base}} = \sqrt{\frac{(\rho_{\text{target}} - 4)(\rho_{\text{target}} - 2)\rho_\infty}{(\rho_\infty - 4)(\rho_\infty - 2)\rho_{\text{target}}}}$      $\triangleright$ Base rectification coefficient

15: $r_t^{\text{ACR}} = r_{\text{base}} \cdot (1 + \lambda\gamma_t)$         $\triangleright$ Final ACR coefficient

16: **if** $\rho_{\text{target}} > 4$ **then**

17:    $\boldsymbol{\theta}^{(t)} = \boldsymbol{\theta}^{(t-1)} - \eta \cdot \text{clip}\left(r_t^{\text{ACR}} \cdot \frac{\hat{\mathbf{m}}^{(t)}}{\gamma\hat{\boldsymbol{\nu}}^{(t)} + \epsilon}\right)$     $\triangleright$ Rectified and clipped update

18: **else**

19:    $\boldsymbol{\theta}^{(t)} = \boldsymbol{\theta}^{(t-1)} - \eta \cdot \text{clip}\left(\frac{\hat{\mathbf{m}}^{(t)}}{\gamma\hat{\boldsymbol{\nu}}^{(t)} + \epsilon}\right)$      $\triangleright$ Un-rectified and clipped update

20: **end if**

---

### E.1 PREDICTION SOLUTIONS SAMPLES

In figure 7 we report visualization of the predicted PDE solutions using our method ACR.

Figure 7: Predictions of PDE datasets using ACR.

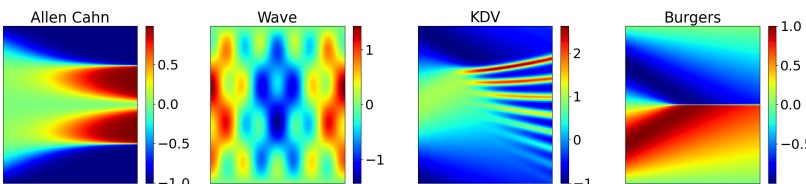

## F ADDITIONAL RELATED WORKS

**Adaptive momentum-based first order methods.** Adam (Kingma & Ba, 2015) remains the most widely used momentum-based optimization algorithm, leveraging adaptive learning rates and exponential moving averages of gradients. RAdam (Liu et al., 2020) addresses the high variance of Adam's adaptive learning rate by introducing a rectification mechanism that stabilizes training in the early phases. AdEMAMix (Pagliardini et al., 2025) extends this family by incorporating a secondary exponential moving average of gradients, enabling the optimizer to retain information from older gradient directions. While our approach also builds upon momentum, it differs fundamentally by operating within the quasi-Newton family of second-order methods. Specifically, it employs layer-wise Hessian approximations to enhance curvature information for improved optimization dynamics.

**Second-Order Optimization for LLMs.** Sivan et al. (2024) propose decomposing the gradient into two orthogonal components, applying Newton's method to one and Adam to the other. Gupta et al. (2018) approximate the empirical Fisher matrix by maintaining two square preconditioners, which are combined with the gradient using a Kronecker product. Building on this, Vyas et al. (2025) further project the preconditioners into the eigenspace to improve efficiency. Liu et al. (2024a) approximate the diagonal of the Gauss-Newton matrix for scalable preconditioning. In contrast to these approaches, we propose rescaling the adaptive learning rate to mitigate its high variance during the early stages of training.

**Functional PINN Optimizers.** Müller & Zeinhofer (2023) propose the use of the Energy Natural Gradient (ENG) method that is equivalent to a newton method in function space. Dangel et al. (2024) adapt the ENG method with a layerwise structure. Schwencke & Furtlehner (2025) explore optimization in the function space induced by the neural tangent kernel (NTK), and Jnini & Vella (2024) develop a Hessian-free optimization method in functional space. Additionally, Jnini et al. (2024) propose a Gauss-Newton method tailored for functional optimization in PINNs. A comprehensive overview of functional optimization approaches in scientific machine learning is presented by Müller & Zeinhofer (2024). Our approach is related in that it approximates a second-order matrix, yet it differs in two important aspects: (1) it supports minibatch training, and (2) it is applicable in a layerwise setting.

**PINN Methods with Eigenvalues.** Wang et al. (2022) leverage the eigenvalues of the Neural Tangent Kernel (NTK) to mitigate disparities in the convergence rates of different PINN losses. Schwencke & Furtlehner (2025) compute eigenvalues from concatenated domain and boundary features, while Jnini & Vella (2024) extract eigenvalues from the Gauss-Newton (GN) matrix. Wang et al. (2025) introduce SOAP, an approximation of the GN matrix that computes eigenpairs via the power iteration algorithm. Our method similarly projects gradients and exponential moving averages (EMA) onto the eigenvectors of the Hessian, but differs by estimating these eigenvectors iteratively using a single round of power iteration.

**Other Optimizers for PINNs.** Rathore et al. (2024) propose a sequential optimization strategy for PINNs, combining Adam, L-BFGS, and a Nyström-accelerated Newton conjugate gradient method (NNCG). Hao et al. (2024) introduce a Gauss-Newton method tailored for solving PDEs with neural networks. Yao et al. (2023) suggest applying separate Adam optimizers to each component of the

PINN loss function, such as the boundary and initial conditions. Liu et al. (2024b) leverage multitask learning to address gradient imbalances between initial/boundary conditions and data terms. Similarly, Hwang & Lim (2024) propose constraining gradients to lie within a dual cone region to improve training dynamics.

## G  REBUTTAL: ADDITIONAL EXPERIMENTAL RESULTS

All experiments are on Wave PDE dataset unless we write otherwise.

### G.1  ABLATION OF INTERNAL COMPONENTS OF ACR

To clarify the respective contributions of the internal mechanisms of Adaptive Confidence Rectification (ACR), we decompose it into three components and study their individual effects. Specifically, ACR consists of:
(i) variance-based confidence estimation ($\beta_s = 1.0$),
(ii) conservative safe fallback ($\rho_{\text{safe}} = 10^6$), and
(iii) gradient-aware scaling ($\lambda = 0$ when disabled).
A configuration without these mechanisms approximates standard RAdam.

We conduct two complementary studies: removal of single components (Study 1) and isolation of individual components (Study 2). All results are reported on the L2 reconstruction error ($\times 10^{-6}$).

**Removing Single Components.**  Table 8 shows that removing the safe fallback substantially degrades performance, indicating its critical role in stabilizing updates under high uncertainty. Disabling variance tracking or gradient adjustment produces only moderate declines. The full ACR configuration achieves the lowest error.

Table 8: Ablation by removing single components.

| Ablation | Var. Track | Safe FB | Grad. Adj. | L2 Error $\times 10^{-6}$ |
|---|---|---|---|---|
| No Variance Tracking | ✗ | ✓ | ✓ | $3.69 \pm 0.12$ |
| No Safe Fallback | ✓ | ✗ | ✓ | $15.8 \pm 0.45$ |
| No Gradient Adjustment | ✓ | ✓ | ✗ | $3.60 \pm 0.44$ |
| Radam +SOAP | ✗ | ✗ | ✗ | $4.70 \pm 0.60$ |
| ACR +SOAP | ✓ | ✓ | ✓ | $\mathbf{2.71 \pm 0.21}$ |

**Using Single Components in Isolation.**  To assess whether any single mechanism is sufficient, we activate each component independently while disabling the others (Table 9). Variance tracking alone is detrimental due to uncontrolled variance-induced scaling. In contrast, safe fallback and gradient adjustment each provide meaningful improvements, yet still fall short of the full ACR.

Table 9: Ablation by isolating single components.

| Ablation | Var. Track | Safe FB | Grad. Adj. | L2 Error $\times 10^{-6}$ |
|---|---|---|---|---|
| Only Variance Tracking | ✓ | ✗ | ✗ | $10.7 \pm 2.3$ |
| Only Safe Fallback | ✗ | ✓ | ✗ | $3.45 \pm 0.68$ |
| Only Gradient Adjustment | ✗ | ✗ | ✓ | $3.02 \pm 0.37$ |
| ACR +SOAP | ✓ | ✓ | ✓ | $\mathbf{2.71 \pm 0.21}$ |

## G.2  HYPERPARAMETER SENSITIVITY OF ACR

To assess the robustness of Adaptive Confidence Rectification (ACR), we perform a controlled hyperparameter sweep on the Wave benchmark. Each parameter is varied independently while keeping the remaining ones fixed at their default values: $\rho_{\mathrm{safe}} = 10, \quad \tau = 0.1, \quad \lambda = 0.1, \quad \beta_s = 0.99$. This experiment isolates the contribution of each hyperparameter to stability and reconstruction quality. As shown in Table 10, ACR exhibits a broad optimal basin: moderate perturbations in individual parameters do not produce catastrophic degradation, indicating that the method is not overly sensitive to tuning. Notably, the safe fallback scale $\rho_{\mathrm{safe}}$ and the gradient-scaling factor $\lambda$ show the clearest influence on performance, reflecting their roles in uncertainty buffering and update moderation. Meanwhile, the variance smoothing factor $\beta_s$ remains stable near its default value, confirming that variance tracking benefits from conservative temporal filtering.

Table 10: L2 Error for ACR hyperparameters sweep

| Parameter | Values | L2 error |
|---|---|---|
| $\rho_{\mathrm{safe}}$ | 5 | $2.87 \pm 0.63$ |
|  | **10** | **$2.71 \pm 0.21$** |
|  | 20 | $2.91 \pm 0.93$ |
| $\tau$ | 0.05 | $2.81 \pm 0.41$ |
|  | **0.1** | **$2.71 \pm 0.21$** |
|  | 0.2 | $2.88 \pm 0.68$ |
| $\lambda$ | **0.05** | **$2.63 \pm 0.43$** |
|  | 0.1 | $2.71 \pm 0.21$ |
|  | 0.2 | $2.81 \pm 0.27$ |
| $\beta_s$ | 0.95 | $2.86 \pm 0.48$ |
|  | 0.96 | $2.89 \pm 0.56$ |
|  | 0.98 | $2.88 \pm 0.40$ |
|  | **0.99** | **$2.71 \pm 0.21$** |

## G.3  HYPERPARAMETERS SEARCH FOR BETA OF THE BASELINE

We conducted a controlled sweep over the two key optimizer parameters, $\beta_1$ and $\beta_2$, only on the baseline model, using a fixed training budget and identical initialization. For each candidate value of the swept parameter, the remaining hyperparameters were held at default. This allows us to isolate the effect of individual parameters while avoiding confounding interactions. The defaults were set to: $\beta_1 = 0.99 \quad \beta_2 = 0.993$. These defaults were selected because they produced the strongest empirical results in the baseline sweep (see Table 11). Once established, we ran our method using exactly these same defaults, without further tuning. This ensures a fair comparison and supports the claim that our approach requires no additional fine-tuning.

Table 11: Beta Sweep for baseline SOAP+RAdam.

| Parameter | Values | L2 error |
|---|---|---|
| $\beta_1$ | 0.96 | $4.94 \pm 0.95$ |
|  | 0.97 | $4.98 \pm 0.71$ |
|  | 0.98 | $4.81 \pm 0.64$ |
|  | **0.99** | **$4.70 \pm 0.60$** |
|  | 0.995 | $5.18 \pm 0.44$ |
| $\beta_2$ | 0.98 | $4.93 \pm 0.99$ |
|  | 0.99 | $4.95 \pm 0.42$ |
|  | **0.993** | **$4.70 \pm 0.60$** |
|  | 0.995 | $4.83 \pm 0.39$ |

### G.4 HYPERPARAMETER SEARCH FOR LR OF THE BASELINE

A key concern is that the improvements of ACR might stem from an unfavorable learning rate choice for the baseline, rather than from the mechanism itself. To rule this out, we conducted a dedicated learning rate sweep for the baseline optimizer (SOAP+RAdam) without ACR. This isolates the baseline's optimal configuration and ensures a fair comparison.

Table 12 reports the results of a broad learning rate search across four PDE benchmarks. The sweep spans two orders of magnitude and includes commonly used values in prior work. We observe a consistent optimum at **LR = 0.001**, which yields the lowest average L2 error across tasks. All baseline comparisons in the paper therefore use this empirically determined optimal LR.

This result confirms that the baseline operates near its best achievable performance with respect to LR, and thus the improvements introduced by ACR cannot be attributed to a poorly tuned or mismatched baseline learning rate.

Table 12: Learning Rate Sweep for Baseline SOAP+RAdam.

| LR | Allen $\times 10^{-6}$ | Wave $\times 10^{-6}$ | Burgers $\times 10^{-5}$ | KdV $\times 10^{-4}$ |
|---|---|---|---|---|
| 0.0001 | $225 \pm 62$ | $28.39 \pm 1.53$ | $677 \pm 31$ | $7.54 \pm 0.52$ |
| 0.0002 | $99.5 \pm 16$ | $7.76 \pm 0.27$ | $167 \pm 11$ | $6.02 \pm 0.19$ |
| 0.0004 | $91.8 \pm 25$ | $5.62 \pm 0.44$ | $29.0 \pm 4.2$ | $5.69 \pm 0.47$ |
| 0.0008 | $45.6 \pm 6.5$ | $5.24 \pm 0.64$ | $8.83 \pm 1.01$ | $5.65 \pm 0.22$ |
| **0.001** | $\mathbf{41.0 \pm 8.6}$ | $\mathbf{4.70 \pm 0.60}$ | $\mathbf{8.55 \pm 0.49}$ | $\mathbf{5.32 \pm 0.53}$ |
| 0.002 | $48.5 \pm 1.5$ | $4.92 \pm 0.23$ | $9.08 \pm 0.76$ | $5.54 \pm 0.18$ |

### G.5 CONFIDENCE OF SECOND-MOMENT ESTIMATES

To investigate whether the adaptivity of the rectification mechanism continues to influence optimization beyond the initial warm-up behavior of RAdam, we analyze the evolution of the confidence $conf_t$ and the coefficient of variation $CV_t$ of the second-moment estimates throughout training. As shown in Figure 8, $conf_t$ increases steadily over the first several thousand steps and eventually reaches a consistently high level. Meanwhile, $CV_t$ gradually decreases, with some transient fluctuations until approximately step 2000, after which it stabilizes and remains low for the remainder of the observed training (up to step $10,000$).

This sustained combination of high confidence and low variation indicates that the second-moment estimates become highly reliable and remain so throughout training. Notably, this reliability persists well beyond the point where the rectification mechanism of RAdam is expected to have already saturated, suggesting that the adaptivity of the second-moment estimates continues to play a role even after the initial rectification effects are no longer dominant.

Figure 8: Evolution of confidence ($conf_t$) and coefficient of variation ($CV_t$) during training.

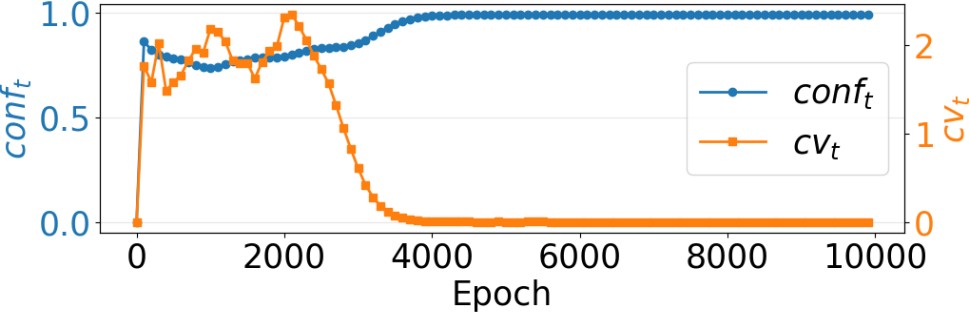

### G.6 DIFFERENCE BETWEEN ACR AND RADAM

In RAdam, the rectification term $\rho_t$ is a *scalar* applied uniformly to all parameters. In contrast, the proposed $\rho_{\text{target}}$ is a *vector* with the same shape as a given layer, enabling per-parameter rescaling and thus a more expressive adaptive mechanism. Figure 9 shows the KDE of $\rho_{\text{target}}$ over 200K training steps for a 256-dimensional layer in the Wave PDE experiment, with the scalar RAdam value $\rho_t$ overlaid. Early in training, the distribution of $\rho_{\text{target}}$ is broad and clearly distinct from $\rho_t$, indicating that the adaptive mechanism captures variability that a single scalar cannot. As optimization proceeds, the distribution narrows and its mean approaches the RAdam value, suggesting that adaptivity is most important in the early phase and naturally diminishes as training stabilizes.

Figure 9: KDE of $\rho_{\text{target}}$ (shaded) vs. scalar $\rho_t$ (red dots) across 200K steps.

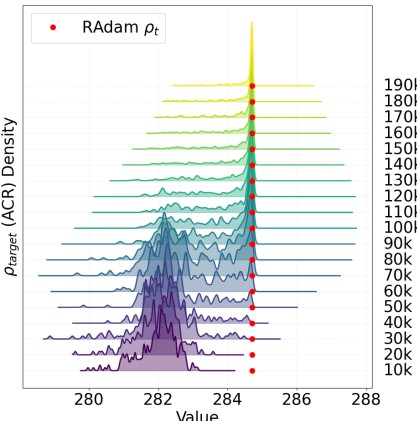

### G.7 COMPARING ACR AND RAW CONFIDENCE SIGNALS

One may wonder whether the adaptive confidence ratio $r_t^{\text{ACR}}$ could be replaced by a simpler heuristic, e.g., $r_t = conf_t$. To test this, we compare the evolution of $r_t^{\text{ACR}}$ and $conf_t$ during training. Figure 10 shows both quantities for a representative linear layer (width 256) on the Wave PDE benchmark. The two curves differ substantially, especially in the early optimization phase where stabilization is most critical, indicating that $r_t^{\text{ACR}}$ is not just a rescaling of $conf_t$. Thus, although heuristic, ACR induces a distinct and empirically useful modulation of the learning rate that is not recovered by simply setting $r_t = conf_t$.

Figure 10: Comparison of the adaptive confidence ratio $r_t^{\text{ACR}}$ and the raw confidence signal $conf_t$.

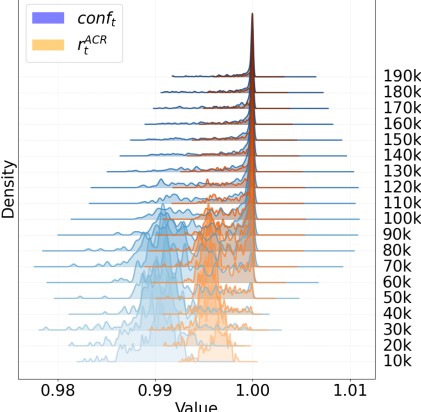

## G.8 RUNTIME AND COMPUTE RESULTS

To assess the practical overhead of ACR, we report wall-clock training times to convergence under matched training budgets for ACR and the SOAP+RAdam baseline. All experiments are run on the same hardware and use identical model architectures, batch sizes, and stopping criteria, so that the number of forward/backward passes is comparable across methods. Table 13 summarizes the wall-clock time to reach convergence for different datasets and model scales. We observe that ACR incurs only a negligible overhead relative to SOAP+RAdam, consistently staying within a small margin across all configurations considered.

Table 13: Wall-clock training times.

| Dataset | Small (70K params) | | Medium (135K params) | | Large (200K params) | |
|---|---|---|---|---|---|---|
| | ACR | RAdam | ACR | RAdam | ACR | RAdam |
| Allen-Cahn | 38min | 30min | 1h 23min | 1h 12min | 1h 41min | 1h 34min |
| Wave | 25min | 21min | 1h 15min | 1h 10min | 1h 29min | 1h 20min |
| KdV | 23min | 20min | 42min | 38min | 53min | 43min |
| Burgers | 32min | 26min | 33min | 27min | 35min | 28min |

## G.9 VALIDATION OF ACR ON FIRST-ORDER OPTIMIZERS

Table 14 presents the performance of our method in the first-order setting. We compare baseline Adam, its rectified variant, and our proposed approach. Across all datasets, ACR consistently yields improved results.

Table 14: MSE performance in first order optimization settings.

| Model | Wave ($\times 10^{-5}$) | Allen ($\times 10^{-4}$) | KdV ($\times 10^{-3}$) | Burgers ($\times 10^{-3}$) |
|---|---|---|---|---|
| RAdam | $4.68 \pm 0.12$ | $10.40 \pm 0.80$ | $2.86 \pm 0.05$ | $3.19 \pm 0.04$ |
| ACR | $3.50 \pm 0.14$ | $8.30 \pm 0.35$ | $2.04 \pm 0.01$ | $1.27 \pm 0.01$ |

### G.10 ALTERNATIVE CONFIDENCE ESTIMATION VIA BOOTSTRAPPING

We additionally evaluate and validate a bootstrap-based confidence estimator as an alternative to our EMA-based approach. Algorithm 6 outlines the bootstrap procedure. Unlike our method (Algorithm 5), which tracks an EMA of squared moment increments to produce the variance term $\mathbf{s}^{(t)}$, the bootstrap variant maintains an ensemble of moment-statistic replicas. At each iteration, each replica is updated independently through a Bernoulli-masked batch resampling step, and the uncertainty is computed as the sample variance across replicas, $\mathbf{u}^{(t)}$. Both methods ultimately yield a CV-based confidence score, but the bootstrap approach is substantially more computationally demanding and, in our experiments, results in slightly lower performance.

---

**Algorithm 5** EMA-based Uncertainty Metric

**Require:** Moment vector $\mathbf{v}^{(t)}$, previous $\mathbf{v}^{(t-1)}$, decay $\beta_{\mathrm{var}}$
**Ensure:** $CV_t$, $conf_t$
    **Initialization**
1: $\mathbf{s}^{(0)} \leftarrow \mathbf{0}$
    **Metric Calculation**
2: $\Delta\mathbf{v}^{(t)} \leftarrow \mathbf{v}^{(t)} - \mathbf{v}^{(t-1)}$
3: $\mathbf{s}^{(t)} \leftarrow \beta_{\mathrm{var}}\,\mathbf{s}^{(t-1)} + (1 - \beta_{\mathrm{var}})\,(\Delta\mathbf{v}^{(t)})^2$
    **Final Calculations**
4: $CV_t \leftarrow \sqrt{\frac{\mathbf{s}^{(t)}}{|\mathbf{v}^{(t)}| + \epsilon}}$
5: $conf_t \leftarrow \frac{1}{1 + CV_t}$

---

**Algorithm 6** Bootstrap-based Uncertainty Metric

**Require:** Ensemble size $N_{\mathrm{boot}}$, vectors $\{\mathbf{v}_j^{(t-1)}\}$, update rule $U$
**Ensure:** $CV_t$, $conf_t$
    **Bootstrap Updates**
1: **for** $j = 1$ to $N_{\mathrm{boot}}$ **do**
2:     Sample batch $\mathcal{B}_j$ with replacement
3:     Compute $\Delta\mathbf{v}_j^{(t)}$ from $\mathcal{B}_j$
4:     $\mathbf{v}_j^{(t)} \leftarrow U(\mathbf{v}_j^{(t-1)}, \Delta\mathbf{v}_j^{(t)})$
5: **end for**
    **Bootstrap Statistics**
6: $\mu^{(t)} \leftarrow \mathrm{Mean}(\{\mathbf{v}_j^{(t)}\})$
7: $\mathbf{u}^{(t)} \leftarrow \mathrm{Var}(\{\mathbf{v}_j^{(t)}\})$
    **Final Calculations**
8: $CV_t \leftarrow \sqrt{\frac{\mathbf{u}^{(t)}}{\mu^{(t)} + \epsilon}}$
9: $conf_t \leftarrow \frac{1}{1 + CV_t}$

---

We benchmark all methods across 10 random seeds, with results summarized in Table 15. The comparison shows that SOAP-ACR adds only negligible computational overhead relative to RAdam, whereas the bootstrap variant consistently incurs higher cost—even with a lightweight implementation and a modest ensemble size ($N = 5$).

Table 15: Bootstrap results for the Wave dataset.

| Model | L2 ($\times 10^{-6}$) | Steptime (seconds) ($\times 10^{-3}$ ↓) | Steps per second (↑) |
|---|---|---|---|
| ACR-Bootstrap | $4.20 \pm 0.32$ | $28.429 \pm 0.227$ | 35.18 |
| RAdam | $4.70 \pm 0.60$ | $25.687 \pm 0.327$ | 38.93 |
| ACR-CV | $2.71 \pm 0.21$ | $25.951 \pm 0.784$ | 38.53 |

# H    REBUTTAL: ADDITIONAL THEORETICAL RESULTS

## H.1    VARIANCE BOUND FOR ACR'S VARIANCE REDUCTION

**Notation.**   We denote gradients by $g_t \in \mathbb{R}^d$. The second-moment exponential moving average (EMA) and its bias-corrected form are

$$v_t = \beta_2 v_{t-1} + (1 - \beta_2)g_t^2, \tag{20}$$

$$\hat{v}_t = \frac{v_t}{1 - \beta_2^t}. \tag{21}$$

The effective per-parameter learning rate is

$$\eta_t^{\text{eff}} = \alpha_t \cdot \frac{r_t}{\sqrt{\hat{v}_t} + \epsilon}, \tag{22}$$

where $r_t$ is the rectification coefficient. RAdam uses $r_t = r(\rho_t)$ with

$$\rho_t = \rho_\infty - \frac{2t\beta_2^t}{1 - \beta_2^t}. \tag{23}$$

ACR modifies this by introducing an adaptive target

$$\rho_{\text{target}} = \text{conf}_t \rho_t + (1 - \text{conf}_t)\rho_{\text{safe}}, \tag{24}$$

where the confidence factor is

$$\text{conf}_t = \frac{1}{1 + CV_t}, \qquad CV_t = \frac{\sqrt{s_t}}{\hat{v}_t + \epsilon}, \qquad s_t = \beta_s s_{t-1} + (1 - \beta_s)(\hat{v}_t - \hat{v}_{t-1})^2. \tag{25}$$

The ACR rectification is then

$$r_t^{\text{ACR}} = r(\rho_{\text{target}})(1 + \lambda\gamma_t), \qquad \gamma_t = \frac{\|\hat{m}_t\|_2}{\|\hat{m}_t\|_2 + \tau\sqrt{d}}. \tag{26}$$

Thus, ACR replaces RAdam's deterministic rectification with a smoothed, confidence-weighted variant that interpolates between an aggressive and a conservative state.

**Assumptions.**   We impose the following standard conditions:

**(A1)** $\rho_t$ is deterministic. Stochasticity arises only from gradient estimates and their effect on $\hat{v}_t$, $\text{conf}_t$, and $\gamma_t$.

**(A2)** $r(\rho)$ is Lipschitz continuous on $[\rho_{\min}, \rho_{\max}] \subset (4, \infty)$ with constant $L_r > 0$.

**(A3)** The gradient-aware factor $\gamma_t$ is bounded: $\gamma_t \in [0, \Gamma]$ and $\text{Var}[\gamma_t] < \infty$.

**(A4)** The variance of $\hat{v}_t$ is finite, i.e., $\text{Var}[\hat{v}_t] < \infty$, and $\sqrt{\hat{v}_t} + \epsilon$ is bounded away from zero.

**Theorem (ACR Variance Reduction).**   Under (A1)–(A4), the effective learning rates for RAdam and ACR are

$$\eta_t^{\text{RAdam}} = \alpha_t \frac{r(\rho_t)}{\sqrt{\hat{v}_t} + \epsilon}, \tag{27}$$

$$\eta_t^{\text{ACR}} = \alpha_t \frac{r(\rho_{\text{target}})(1 + \lambda\gamma_t)}{\sqrt{\hat{v}_t} + \epsilon}. \tag{28}$$

Then the variance of ACR's effective learning rate satisfies

$$\text{Var}[\eta_t^{\text{ACR}}] \leq \text{Var}[\eta_t^{\text{RAdam}}] - \frac{\alpha_t^2 L_r^2}{(\sqrt{\hat{v}_t} + \epsilon)^2}(\rho_t - \rho_{\text{safe}})^2 \text{Var}[\text{conf}_t] + \mathcal{O}\left(\frac{\alpha_t^2}{(\sqrt{\hat{v}_t} + \epsilon)^2}\lambda^2\text{Var}[\gamma_t]\right). \tag{29}$$

The second term on the right-hand side is nonnegative, so ACR reduces the variance of the effective learning rate relative to RAdam by at least

$$\Delta_t = \frac{\alpha_t^2 L_r^2}{(\sqrt{\hat{v}_t} + \epsilon)^2}(\rho_t - \rho_{\text{safe}})^2 \text{Var}[\text{conf}_t]. \tag{30}$$

**Proof sketch.** Write

$$\eta_t^{\mathrm{ACR}} = \eta_0 \, r(\rho_{\mathrm{target}})\big(1 + \lambda\gamma_t\big), \qquad \eta_0 = \frac{\alpha_t}{\sqrt{\hat{v}_t} + \epsilon}. \tag{31}$$

Then

$$\mathrm{Var}[\eta_t^{\mathrm{ACR}}] = \eta_0^2 \, \mathrm{Var}\big[r(\rho_{\mathrm{target}})\big(1 + \lambda\gamma_t\big)\big]. \tag{32}$$

Assuming weak correlation between $\mathrm{conf}_t$ and $\gamma_t$, we use the approximation

$$\mathrm{Var}[XY] \approx \mathbb{E}[X]^2 \mathrm{Var}[Y] + \mathbb{E}[Y]^2 \mathrm{Var}[X]. \tag{33}$$

By Lipschitz continuity of $r(\cdot)$,

$$\big|r(\rho_{\mathrm{target}}) - r(\rho_t)\big| \le L_r \big|\rho_{\mathrm{target}} - \rho_t\big|. \tag{34}$$

Using (24),

$$\rho_{\mathrm{target}} - \rho_t = (1 - \mathrm{conf}_t)(\rho_{\mathrm{safe}} - \rho_t), \tag{35}$$

and hence

$$\mathrm{Var}\big[r(\rho_{\mathrm{target}})\big] \le L_r^2 (\rho_t - \rho_{\mathrm{safe}})^2 \mathrm{Var}[\mathrm{conf}_t]. \tag{36}$$

Combining the above gives

$$\mathrm{Var}[\eta_t^{\mathrm{ACR}}] \le \eta_0^2 \Big( \mathrm{Var}[r(\rho_t)] - L_r^2 (\rho_t - \rho_{\mathrm{safe}})^2 \mathrm{Var}[\mathrm{conf}_t] + \lambda^2 \mathrm{Var}[\gamma_t] \Big), \tag{37}$$

which yields (29). The residual term $\mathcal{O}(\lambda^2 \mathrm{Var}[\gamma_t])$ captures the effect of the gradient-aware factor and remains bounded for small $\lambda$.

**Interpretation.** Equation (29) gives a quantitative guarantee that ACR reduces the variance of the adaptive learning rate. The reduction term $\Delta_t$ scales with both the gap between the aggressive rectification $\rho_t$ and the conservative anchor $\rho_{\mathrm{safe}}$, and the uncertainty of the confidence estimate $\mathrm{Var}[\mathrm{conf}_t]$. When the second-moment estimate is unstable (large $\mathrm{Var}[\mathrm{conf}_t]$), ACR automatically contracts the variance of the rectification, stabilizing early training. As training settles and $\mathrm{Var}[\mathrm{conf}_t] \to 0$, ACR smoothly recovers RAdam's behavior.

**Discussion.** This result links ACR's empirical stability to a formal variance bound on the effective step size. The adaptive confidence mechanism acts as a *variance controller* that interpolates toward a stable rectification anchor. The contraction magnitude $\Delta_t$ quantifies the reduction in stochasticity of parameter updates, showing that the resulting effective learning rate variance is provably smaller than that of RAdam. This helps explain the smoother and more reliable training trajectories observed in our experiments.

