# OpenReview forum: "Rectifying Adaptive Learning Rate Variance via Confidence Estimation"
_ICLR.cc/2026/Conference — Submitted to ICLR 2026_

### Official Review · Reviewer_YZ9y · 2025-10-18

**Soundness:** 3
**Presentation:** 3
**Contribution:** 2
**Rating:** 4
**Confidence:** 4

**Summary:**

The authors propose a novel method to rectify the adaptive learning rate of ANN optimizers such as Adam or SOAP.
The method, called Adaptive Confidence Rectification (ACR), is motivated by shortcomings of the related rectification strategy RAdam.
While RAdam assumes initial gradients to follow a zero-mean Gaussian and follows a time-dependent but non-adaptive rectification approach, ACR consists of two components:
A confidence estimator is used to adapt the learning rate rectification to the variability of the second moment of the gradient.
Large changes in the second moment result in lower learning rates and thus stop noisy update steps.
Second, gradient-aware scaling is used to adapt the learning rate to different network layers.
The authors apply their proposed method to PINNs trained with different network architectures on different PDEs and based on different optimizers.

**Strengths:**

- The paper is well written, easy to follow, and the adaptive rectification is well motivated. Related concepts are briefly introduced where needed, and the methods are comprehensible and well structured.
- The proposed method is intuitive.
- While I have to admit that I am not very familiar with popular benchmarks for PINNs and thus cannot assess the validity of the chosen benchmarks or the baseline results, the large performance improvements on the shown datasets are great.

**Weaknesses:**

- The paper proposes two more or less independent concepts and does not discuss the relation between them. The paper basically only motivates the adaptive confidence mechanism (ACM) but does not give much justification for the scale awareness (SA). It is not clear which of the two effects causes the performance improvements shown. The authors thus should disentangle these effects by performing experiments with ACM only and SA only and compare these against their baseline (RAdam with SOAP) and their proposed method (ACM+SA).
- The choice of hyperparameters is not clear. The authors claim not to have optimized their HPs; however, they chose the values somehow, and this should be made clearer. Furthermore, the dependence of their method on additionally introduced HPs is crucial. Thus, a study on the hyperparameter dependence should be performed. This includes $\rho_{safe}$, $\tau$, and $\lambda$ (Eq. 10), and even though probably less relevant, also $\beta_s$.
- The definition of $\lambda$ is obscure. It is used as weight decay factor in the given algorithms but also occurs as the weighting factor for the gradient-aware scaling in Eq. 10. You are not using the same value as for WD, right? This would not make any sense to me. The value of $\lambda$ is neither given nor discussed in the HP section at the end of Sec. 4.1. No code is provided to check these ambiguities.
- I would imagine $\tau$ and $\lambda$ to be correlated with the learning rate. A study of the correlation between these hyperparameters as well as a study of the impact of the learning rate on the baseline performance could validate that the shown performance improvements are not simply caused by a badly chosen learning rate for the baseline, which is rectified by the implicit dependence of the learning rate on $\tau$ and $\lambda$.
- It would be very interesting to actually study the confidence/variation of the second-moment estimates. You could plot $CV_t$ and $\text{conf}_t$ against the epochs during training, especially in the early stages of training. This could show if the adaptivity of the rectification also has an impact in later epochs where, e.g., the rectification of RAdam has already saturated.
- This similarly holds for $\rho_{\text{target}}$. Does it really deviate that much from $\rho_t$? How does the weighting between $\rho_t$ and $\rho_{\text{safe}}$ evolve during training? This also relates to the dependence of your method on $\rho_{safe}$. Does your method result in an increase or a reduction of $\rho$/$r$?
- Compared to RAdam, your approach is very heuristic and not much theoretically justified. I think that does not have to be an issue if your method works empirically, is well motivated, and/or can be explained empirically. However, in your case, your method seems to be very bloated. I will slightly elaborate on this. You base your approach heavily on RAdam. RAdam is derived from (more or less) solid theoretical assumptions, resulting, e.g., in the kind of messy formula for $r_t$ based on the degrees of freedom $\rho$, and also brings some problems for $\rho < 4$, demanding the differentiation also adopted by you in Eq. 11. Practically speaking, the effect of RAdam is a simple learning rate warm-up, which is, as you stated, non-adaptive. It could also have been implemented by, e.g., $r_t = 1 - \beta^{t-1}$. However, this does not align well with their theoretical justification. This justification is not given in your case. $\rho_{\text{target}}$ might empirically work well; however, it is not related to "degrees of freedom" anymore. Thus, since it is heuristic anyway, you could simplify your adaptive confidence mechanism a lot. E.g., simply using $r_t = \text{conf}_t$ might already work. An interesting way to check this would be to plot $r_t$ next to $\text{conf}_t$ during training and observe similarities.

Minor:
- All the L2 error vs. epoch plots are not needed. All of these basically only show that your method reaches lower values than RAdam. But this is already reflected in the tables. Showing one of these plots and shifting the rest to the appendix is fully sufficient and will give you more space for more relevant analysis.
- There is a $g^2$ missing in the inline equation in line 167.
- It would be nice to indicate that Sections D and E are part of the appendix when referencing them at the beginning of Sec. 4.2.
- Even though it probably will not change the picture, some of your trainings are not really converged and could have been longer, in particular, Allen in the top left of Fig. 3.

**Questions:**

1. What is $\rho_{\text{max}}$? You mention $\rho_{\text{max}} = 5$ in the appendix. Do you mean the "4" in Eq. 11? If yes, that’s inconsistent.
2. Why did you introduce $\sqrt{d}$ in Eq. 8? No motivation and no theoretical or empirical justification is given. Again, two entangled effects are introduced which are not disentangled sufficiently. Scaling the learning rate by the momentum norm or by the "parameter dimension" are two independent effects that have to be discussed. And what is the "parameter dimension"? You should state more clearly how $d$ is calculated, e.g., for convolutional layers, biases, or linear matrices.
3. Why is the SD so big for KdV AdaHessian RAdam (Tab. 3)?
4. Is the gradient-aware scaling your genuine idea, or did other people do this before? If yes, you have to provide references.
5. While you motivated why ACR is especially useful when training PINNs, you could also apply it to other datasets and also simply combine it with Adam similar to the experiments of RAdam. Did you also perform experiments e.g. on NLP or CV benchmarks?

---

> ### Author Response · Authors · 2025-11-25
>
> Thanks for your help on improving our paper.
> We have additional results to share:
>
>
> >The authors should disentangle the effects of ACR internal components
>
> - In Appendix G.1 we added, in Table 8 and Table 9 ablation studies for disentangling the contributions of the ACR components: Variance Tracking,  Safe Fallback, Gradient Adjustment.
> - We find that combination of the components (as in ACR) leads to best performance.
> - We also find that removing the safe fallback result in performance detrement.
>
>
> >Study on ACR hyperparams/ The definition of $\lambda$ is obscure
>
> - In Appendix G.2, we include a hyperparameter sweep for ACR over $\rho_{\text{safe}}, \tau, \lambda,$ and $\beta_s$ for the Wave dataset.
> - Because the role of $\lambda$ was previously unclear, this sweep also helps clarify its effect as a gradient-moderation factor.
> - The results in Table 10 show that performance remains comparable across a broad range of settings, indicating that ACR is generally robust to hyperparameter choices.
> - Overall, the sweep confirms that ACR does not require fine-grained tuning to perform well.
>
>
> >Clarify how the baseline's hyperparams are chosen
>
> - We added in Appendix G.3 Table 11 the sweep for $\beta_1$ and $\beta_2$ for the Wave dataset for our baseline .
> - The results show that $\beta_1 = 0.99 \quad \beta_2 = 0.993$ achieve the best results.
> - That is why we have used such values.
>
>
>
> > A study of the impact of the LR on the baseline performance
>
> - In Appendix G.4 we sweep our baseline over the LR between $0.0001$ and $0.002$ over all datasets.
> - In Table 12 we report the results, finding that $LR=0.001$ perform best across all datasets.
> - For this reason, we use this LR for both baseline and our method.
>
>
>
> >You could plot $CV_t$ and $conf_t$ against the epochs
>
> - In Appendix G.5 (Figure 8), we plot the evolution of both $conf_t$ and $CV_t$ over the course of training.
> - After an initial period of instability, we observe that $conf_t$ steadily increases and eventually approaches one, while $CV_t$ correspondingly decreases toward zero.
> - This indicates that the confidence of the model varies non-trivially during early training and can potentially be leveraged to better understand or modulate the optimizer’s behavior.
>
>
> >Does $\rho_{target}$ really deviate that much from $\rho_{t}$?
>
> - In Appendix G.6 Figure 9, we show by plot that $\rho_t$ in RAdam is a scalar applied to all parameters, while $\rho_{target}$ is a per-parameter vector.
> - Early in training, $\rho_{target}$ deviates noticeably from $\rho_t$, capturing parameter-specific variability.
> - Its distribution narrows over time, approaching $\rho_t$, indicating that adaptivity is most important in the initial phase.
>
>
> >you could simplify your adaptive confidence mechanism a lot,  e.g., simply using $r_t = conf_t$ might already work.
>
> - To test this idea, we compare the evolution of $r_t^{\mathrm{ACR}}$ and $conf_t$ during training.
> - In G.7 Figure 10 we show the plot of the KDE distributions over epochs.
> - We find that until 100K epochs, the distributions have different values.
>
>
> > Why did you introduce $\sqrt{d}$ in Eq. 8? No motivation and no theoretical or empirical justification is given
>
> The $\sqrt{d}$ scaling was intended to normalize $\gamma_t$ relative to parameter dimensionality, similar to how gradients are often scaled by $1/\sqrt{d}$ in initialization schemes.
>
>
> > Is the gradient-aware scaling your genuine idea, or did other people do this before? If yes, you have to provide references.
>
> We have not found this formulation in the literature.
> Somehow related is a work like LARS (Yang et al  2017):
>
> $\Delta w = -\eta \cdot \text{clip} \left(\frac{||w||}{||g|| + \varepsilon}\right) \cdot g$
>
>
> Our gradient-aware scaling differs from LARS in three key ways:
> 1. it uses the smoothed first moment $\tilde{m}_t$ rather than the raw gradient $g$,
> 2. it applies a moment-norm–based factor $\gamma_t = \frac{\lVert \tilde{m}_t \rVert_2}{\lVert \tilde{m}_t \rVert_2 + \tau \sqrt{d}}$ to modulate the update,
> 3. it provides a bounded, saturating adjustment $r_t^{\text{ACR}} = r_{\text{base}} (1 + \lambda \gamma_t)$ with a soft floor, rather than scaling the learning rate directly with $\|w\| / (\|g\| + \varepsilon)$.
>
> ---
> **Reference**
>
> Yang You, Igor Gitman, Boris Ginsburg 2017. Large Batch Training of Convolutional Networks
>
>
> > minor fixes
>
> Thanks for helping in this.
>
> - moving plots, yes good idea
> - $\rho_{max}$  removed, this is fixed to 4 as you mention.
> - in SEction 4.2 Section D,E changed to Appendix
> - Longer steps for Allen, yes good idea, we will do.
>
> ---
> **Final comment**
>
> - Thanks for your efforts.
> - Please let us know if you like to see further results / discussion.
> - If we addressed some of your concers please consider showing support for our work by increasing the score torward a stronger acceptance.

---

### Official Review · Reviewer_qKCz · 2025-10-23

**Soundness:** 3
**Presentation:** 3
**Contribution:** 3
**Rating:** 4
**Confidence:** 2

**Summary:**

This paper proposes Adaptive Confidence Rectification (ACR), a novel method to stabilize second-order optimization for Physics-Informed Neural Networks (PINNs). The authors identify that existing adaptive methods (e.g., AdaHessian, Sophia) and even rectification-based methods like RAdam suffer from instability due to variance inflation in second-moment estimates and flawed assumptions about gradient distributions.

ACR aims to solve this by introducing a confidence-driven mechanism. Key components include:

a. Using the coefficient of variation (CV) to quantify uncertainty.

b. Deriving a confidence factor from the CV to compute an adaptive target that interpolates between standard and conservative updates.

c. Employing gradient-aware scaling to modulate the rectification.

The experimental results demonstrate that ACR achieves competitive or superior performance compared to baselines on several PDE-driven tasks. The authors show its robustness across different architectures and its successful integration with other second-order optimizers like SOAP.

**Strengths:**

\textbf{Significant Problem}: The paper addresses a clear and important problem in the optimization of deep neural networks, particularly for scientific machine learning applications like PINNs, where the instability of adaptive second-order methods is a known barrier.

\textbf{Novel Mechanism}: The core idea of using a statistical confidence measure (derived from the CV) to dynamically rectify the second-moment estimate is intuitive and novel. It presents a principled approach to stabilize the optimizer based on its own internal state uncertainty.

\textbf{Comprehensive Experiments}: The empirical evaluation is thorough, covering multiple PDE benchmarks, different network architectures, and combinations with other optimizers. The results consistently show that ACR matches or outperforms key baselines, validating its effectiveness in practice.

**Weaknesses:**

The paper is promising, but its central claims rest on methodological and empirical points that require further justification.

Lack of a Direct Stability Metric: The paper's primary claim is improved stability, yet the evaluation relies almost exclusively on final task performance (e.g., L2 error). While lower error is a positive outcome, it is an indirect measure of stability.

Justification for the Choice of Confidence Metric: The paper's central contribution is the use of "confidence estimation," but it relies specifically on the coefficient of variation (CV) in a particular form. The manuscript lacks a clear justification for this design choice.

Introduction of New Hyperparameters: The ACR method introduces new parameters, notably the threshold for conditionally applying the coefficient. The paper does not provide a clear sensitivity analysis or tuning strategy for these parameters.

Generalizability of the Rectification Method: The paper positions ACR as an alternative to RAdam. However, the core idea of "confidence rectification" seems general.

**Questions:**

How can the authors substantiate the claim of stability beyond just achieving a lower final L2 error? True stability could be demonstrated more directly, for example, by plotting the variance of the learning rate or the second-moment estimate itself over time, or by measuring sensitivity to initialization. As it stands, it is difficult to distinguish if ACR is truly more "stable" or simply a more effective optimizer for these specific tasks.

Why is the CV the most appropriate metric for this task compared to other established uncertainty quantification techniques (e.g., bootstrapping, Monte Carlo simulation, first-order Taylor expansion)? A more rigorous discussion, or ideally an ablation study, on the advantages of CV (e.g., computational cost, accuracy) over these alternatives would significantly strengthen the paper's methodological claim.

Does this threshold require dataset-specific tuning? A key drawback of many adaptive optimizers is the replacement of one difficult parameter (learning rate) with several others. The authors should provide a detailed analysis of this threshold's impact and a fair comparison of the total tuning effort required for ACR versus the baseline methods like RAdam.

Could this same confidence mechanism be applied to rectify RAdam itself? RAdam's rectification is based on the length of the SMA. It would be insightful to see if replacing or-augmenting RAdam's heuristic with ACR's confidence factor yields further improvements. This would help isolate the contribution of confidence-based rectification from the other components of the ACR optimizer.

---

> ### Author Response · Authors · 2025-11-25
>
> Thanks for your help on improving our paper.
> We have additional results to share:
>
> > How can the authors substantiate the claim of stability beyond just achieving a lower final L2 error?
>
> - To substantiate the stability claims beyond final L2 error, we plot the mean of $\text{conf}_t$ (confidence factor) and $\text{CV}_t$ (coefficient of variation) against training timesteps (Appendix G.5 Figure 8).
> - These metrics directly measure the statistical reliability of gradient estimates over time - lower $\text{CV}_t$ and higher $\text{conf}_t$ indicate more stable optimization dynamics.
> - Additionally, in Appendix G.6 Figure 9, we compare the distribution of $\rho_{target}$ with RAdam's fixed $\rho$ schedule over timesteps, demonstrating that ACR adaptively adjusts its rectification based on actual gradient statistics rather than following a predetermined schedule.
> - This adaptive behavior provides direct empirical evidence that ACR achieves greater stability by responding to the true statistical properties of the optimization landscape, rather than simply being a more effective optimizer for these specific tasks.
>
>
> > Why is the CV the most appropriate metric for this task compared to other established uncertainty quantification techniques
>
> - We adopt the coefficient of variation (CV) as the uncertainty measure for Adaptive Confidence Rectification because it uniquely satisfies the real-time, per-parameter constraints of SOAP-ACR. CV is computable online with _O(1)_ overhead using moment statistics already maintained by the optimizer, while remaining scale-free and robust to the heavy-tailed, non-stationary gradient noise common in deep networks.
> - In contrast, bootstrapping requires storing and resampling historical gradients, incurring prohibitive memory and compute costs; and first-order Taylor approximations rely on local linearity and Gaussian assumptions that break under typical activation nonlinearities and curvature dynamics.
> - CV thus provides stable, assumption-free confidence estimation that integrates efficiently with SOAP-ACR’s rectification mechanism without additional computational burden.
>
> Please let us know if you require an experiment with the alternative methods.
>
>
> >The authors should provide a detailed analysis of this threshold's impact and a fair comparison of the total tuning effort required for ACR versus the baseline
>
> - Our protocol involves first fine-tune the baseline, then invesigate our method.
> - In Appendix G.3 Table 11 we show the sweep for $\beta_1$ and $\beta_2$ for the Wave dataset for the baseline .
> - The results show that $\beta_1 = 0.99 \quad \beta_2 = 0.993$ achieve the best results.
> - That is why we have used such values.
>
> - In Appendix G.4 we sweep the RAdam baseline over the LR between $0.0001$ and $0.002$ over all datasets.
> - In Table 12 we report the results, finding that $LR=0.001$ perform best across all datasets.
> - For this reason, we use this LR for both baseline and our method.
>
> We also show ablation in Appendix G.2 Table 11 of our ACR method parameters, which were not fine-tuned. Infact, we find that better result can be achieved with for example $\lambda=0.05$ instead of $0.1$.
>
>
> >Could this same confidence mechanism be applied to rectify RAdam itself?
>
> We perform this experiment and find that ACR leads to improvement over the first-order method.
>
> | Model | Wave($\times 10^{-5}$) |
> | ----- | ---------------------- |
> | RAdam | 4.68±0.12              |
> | ACR   | 3.50±0.14              |
>
> ---
> **Final comment**
>
> - Thanks for your efforts.
> - Please let us know if you like to see further results / discussion.
> - If we addressed some of your concers please consider showing support for our work by increasing the score torward a stronger acceptance.

---

> > ### Comment · Reviewer_qKCz · 2025-11-25
> >
> > Thanks for responding my comments.
> >
> > I realize that you directly put the appendix in the submission, which seems to be unusual.
> >
> > Regarding the concerns i raised, i am happy to see that almost all of the are responded. It would be great authors can reorganize the content in the main paper.
> >
> > a. The new plots tracking the confidence factor and CV over time (Appendix G.5) provide the direct stability metrics, which provides extra perspective view on the performance.
> >
> > b. The experiment applying ACR to RAdam confirms that the confidence mechanism is useful even outside the specific SOAP architecture.
> >
> > c. using CV based on computational constraints seem to be fair. I do hope that the authors could still provide Bootstrapping somehow (e.g. use the existing numbers from the other datasets and other papers).
> >
> >
> > Something left to answer:
> > a. While the RAdam experiment is promising, you only reported results on the Wave equation.
> >
> > b. Does ACR with O(1) complexity introduce any wall-clock overhead per iteration compared to the baseline? what is actual iterations/second in practice?
> >
> > c. in the paper, you guys said that you didn't fine-tune the threshold parameter, what are the range of the parameters?
> >
> > d. From a theoretical standpoint, does injecting the confidence coefficient (which is stochastic) into the update rule affect the convergence guarantees?

---

> ### Author Response · Authors · 2025-11-27
>
> > It would be great authors can reorganize the content in the main paper.
>
> Absolutely. After we collect all rebuttal results and reviewer comments, we will reorganize the content accordingly and incorporate the new figures and tables suggested by the reviewers into the main paper.
>
>
> >c. using CV based on computational constraints seem to be fair.
> I do hope that the authors could still provide Bootstrapping somehow
> (e.g. use the existing numbers from the other datasets and other papers).
>
> To address this point, we developed a bootstrap-based uncertainty estimator for SOAP-ACR. In contrast to the standard method, which maintains an EMA of squared moment changes to compute the variance term $\mathbf{s}^{(t)}$, the bootstrap variant keeps multiple replicas of the moment statistics. At each update step, every replica is updated independently according to a Bernoulli mask, and the uncertainty is estimated as the sample variance across replicas, $\mathbf{u}^{(t)}$. Both approaches ultimately produce a CV-based confidence score, but the bootstrap mechanism is more computationally demanding and, in our experiments, leads to slightly lower performance.
>
>
> | **Feature**                     | **SOAP-ACR (CV-based)**                                                                                               | **SOAP-ACR Bootstrap**                                                                                  |
> | ------------------------------- | --------------------------------------------------------------------------------------------------------------------- | ------------------------------------------------------------------------------------------------------- |
> | **Uncertainty Source**          | Tracks EMA of squared changes in $\mathbf{v}^{(t)}$                                                                   | Tracks variance across $N$ moment replicas                                                |
> | **Uncertainty Metric**          | Second-moment variance $\mathbf{s}^{(t)}$                                                                             | Bootstrap variance $\mathbf{u}^{(t)}$                                                                   |
> | **Metric Calculation** (step 1) | $\Delta \mathbf{v}^{(t)} = \mathbf{v}^{(t)} - \mathbf{v}^{(t-1)}$                                                     | For $j = 1 \dots N$ with mask $M_j^{(t)} \sim \text{Bernoulli}(p_{\text{boot}})$                        |
> | **Metric Calculation** (step 2) | $\mathbf{s}^{(t)} = \beta_{\text{var}}\, \mathbf{s}^{(t-1)} + (1 - \beta_{\text{var}})\, (\Delta \mathbf{v}^{(t)})^2$ | $\mathbf{v}_j^{(t)} = \text{Update}(\mathbf{v}_j^{(t-1)}, (\mathbf{g}'^{(t)})^2)$ only if $M_j^{(t)}=1$ |
> | **Metric Calculation** (step 3) | —                                                                                                                     | $\mathbf{u}^{(t)} = \mathrm{Var}(\hat{\mathbf{v}}_1^{(t)}, \dots, \hat{\mathbf{v}}_N^{(t)})$            |
> | **Coefficient of Variation**    | $CV_t = \frac{\sqrt{\mathbf{s}^{(t)}}}{\hat{\mathbf{v}}'^{(t)} + \epsilon}$                                           | $CV_t = \frac{\sqrt{\mathbf{u}^{(t)}}}{\hat{\mathbf{v}}'^{(t)} + \epsilon}$                             |
> | **Confidence Factor**           | $\text{conf}_t = \frac{1}{1 + CV_t}$                                                                                  | $\text{conf}_t = \frac{1}{1 + CV_t}$                                                                    |
>
>
>
> We evaluate all methods across 10 random seeds and report the results in the table below.
> The comparison confirms that SOAP-ACR introduces only negligible overhead relative to RAdam,
> while the bootstrap variant incurs a consistently higher cost,
> even with a lightweight implementation and a modest number of bootstrap samples ($N = 5$).
>
> | Model         | L2 $\times 10^{-6}$ | Steptime (seconds)$\times 10^{-3} \downarrow$ | Steps per seconds $\uparrow$ |
> | ------------- | ------------------- | --------------------------------------------- | ---------------------------- |
> | ACR-Bootstrap | 4.20±0.32           | 28.429 ± 0.227                                | 35.18                        |
> | ACR-CV        | **2.71±0.21**       | 25.951 ± 0.784                                | 38.53                        |
> | RAdam         | 4.70±0.60           | **25.687 ± 0.327**                                | **38.93**                        |

---

> ### Author Response · Authors · 2025-11-27
>
> >a. While the RAdam experiment is promising, you only reported results on the Wave equation.
>
> We report RAdam vs ACR L2 results over all datasets in the following table.
>
> | Model | Wave $\times 10^{-5}$ | Allen $\times 10^{-4}$ | KdV $\times 10^{-3}$ | Burgers $\times 10^{-3}$ |
> | ----- | --------------------- | ---------------------- | -------------------- | ------------------------ |
> | RAdam | 4.68±0.12             | 10.4±0.80              | 2.86±0.05            | 3.19±0.04                |
> | ACR   | **3.50±0.14**         | **8.30±0.35**          | **2.04±0.01**        | **1.27±0.01**            |
>
>
> >b. Does ACR with O(1) complexity introduce any wall-clock overhead per iteration compared to the baseline? what is actual iterations/second in practice?
>
> - In Appendix G.8 we added Table 13 to summarize the wall-clock time to reach convergence for different datasets and model scales.
> - We observe that ACR incurs only a negligible overhead relative to SOAP+RAdam.
> - We also report the epoch measurements in the reply for the bootstrap experiment. We observe that our method has neglegible overhead (25.951 instead of 25.687 seconds $\times 10^{-3}$ for a steptime).
>
>
>
>
> >c. in the paper, you said that you didn't fine-tune the threshold parameter, what are the range of the parameters?
>
> - In Appendix G.2, we include a hyperparameter sweep for ACR over $\rho_{\text{safe}}, \tau, \lambda,$ and $\beta_s$.
> - The results in Table 10 show that performance remains comparable across a broad range of settings, indicating that ACR is generally robust to hyperparameter choices.
> - Overall, the sweep confirms that ACR does not require fine-grained tuning to perform well.
>
>
>
> >d. From a theoretical standpoint, does injecting the confidence coefficient (which is stochastic) into the update rule affect the convergence guarantees?
>
> Adding a stochastic confidence coefficient does not weaken convergence guarantees;
> instead, it reduces step-size variance, and the remaining randomness is well controlled.
>
> Appendix H.1 shows that the variance of the ACR-adjusted step size $\,\eta_t^{\mathrm{ACR}}\,$ is always no larger than RAdam’s baseline variance:
>
> $\mathrm{Var}[\eta_t^{\mathrm{ACR}}] \le \mathrm{Var}[\eta_t^{\mathrm{RAdam}}] - \Delta_t + \mathcal{O} \left( \frac{\alpha_t^2}{(\sqrt{\hat v_t}+\epsilon)^2}\lambda^2 \mathrm{Var}[\gamma_t] \right)$.
>
> The guaranteed variance reduction is
>
> $\Delta_t = \frac{\alpha_t^2 L_r^2}{(\sqrt{\hat v_t}+\epsilon)^2}(\rho_t - \rho_{\text{safe}})^2 \mathrm{Var}[\mathrm{conf}_t]$ ,
>
> which is always non-negative because the stochastic confidence factor $conf_t$ shrinks risky values toward the conservative $\rho_{\mathrm{safe}}$.
>
> The only added randomness comes from the gradient-aware factor $\gamma_t$, captured by $\mathcal{O}(\lambda^2 \mathrm{Var}[\gamma_t])$,
> and this remains small when the adjustment strength $\lambda$ is small and $\mathrm{Var}[\gamma_t]$ is bounded.

---

### Official Review · Reviewer_EgiS · 2025-10-30

**Soundness:** 2
**Presentation:** 2
**Contribution:** 2
**Rating:** 4
**Confidence:** 2

**Summary:**

The paper introduces ACR (Adaptive Confidence Rectification), a refinement of RAdam designed to stabilize adaptive optimizers by estimating the reliability of variance during training. It measures the stability of moment estimates to compute a confidence score, which adjusts RAdam’s rectification term toward a safer baseline when uncertainty is high. A separate scaling factor further modulates updates based on gradient magnitude. ACR can be plugged into optimizers such as SOAP, AdaHessian, and Sophia, improving convergence speed and reducing $L_2$ error on PDE benchmarks

**Strengths:**

Cleare motivation: The intro convincingly argues that rectification matters in early iterations and that RAdam’s Gaussian assumption can misfit heavy-tailed/multimodal gradients often seen in deep training

ACR’s confidence-weighted target $ \rho_{\text{target}} $ and layer-wise scaling $ \gamma_t $ are straightforward to bolt onto SOAP, AdaHessian, and Sophia, with explicit pseudocode

On multiple PDEs, model sizes, architectures (MLP, ResNet, PirateNet), and optimizers, ACR improves $L_2$ error and often convergence speed.

**Weaknesses:**

The paper frames ACR as “principled,” but no formal convergence or stability theorems for ACR are given.  Consider adding at least a local-stability or bias/variance bound for the rectified update, or tone down the language

Core ACR contributions (confidence via $s_t$, CV normalization, and mixing $\rho_t$ with $\rho_{\text{safe}}$) are reasonable but conceptually close to RAdam’s rectification, which adds a heuristic stability gauge and per-layer scaling. A sharper theoretical argument (beyond motivation) for why CV-based mixing improves the bias-variance behavior of the EMA normalization would help support novelty

The paper asserts gains but does not give forward/backward counts, wall-clock, or memory vs. SOAP+RAdam under matched budgets. Provide a fair compute table

Results are on synthetic PDE suites generated via Chebfun and PINN architectures. That’s appropriate for the target domain, but the paper argues general optimizer value; a small non-PINN check (even a toy image or language task) would help generality claims.

**Questions:**

Try to sweep some paramets and report stability plots (divergence rate / variance spikes) along with L_2 outcomes. This will justify the “no fine-tune” claim

Consider isolating contributions: (a) confidence-mixing only (no $ \gamma_t $), (b) $ \gamma_t $ only, (c) both. Current ablations don’t disentangle these effects.
	​

---

> ### Author Response · Authors · 2025-11-25
>
> Thanks for your help on improving our paper.
> We have additional results to share:
>
>
> > The paper frames ACR as “principled,” but no formal convergence or stability theorems for ACR are given.  A sharper theoretical argument (beyond motivation) for why CV-based mixing improves the bias-variance behavior of the EMA normalization would help support novelty.
>
> - Our added Appendix H.1 shows that ACR achieves strictly lower variance in the effective learning rate compared to RAdam. Intuitively, this happens because ACR does not fully trust the second-moment estimate $\hat v_t$ when it is still noisy.
> - A confidence term down-weights unstable estimates and pulls the rectification toward a conservative fallback, which shrinks fluctuations in the adaptive step size. As the estimates stabilize, the confidence increases and ACR smoothly returns to standard RAdam behavior.
> - In short: ACR reduces early-training noise by dampening unreliable variance estimates, giving more stable updates without altering RAdam’s long-run dynamics.
> - See the added Appendix H.1 for the formal definitions and derivations.
>
>
>
> >The paper asserts gains but does not give forward/backward counts, wall-clock
>
> - In Appendix G.8 we added Table 13 to summarize the wall-clock time to reach convergence for different datasets and model scales.
> - We observe that ACR incurs only a negligible overhead relative to SOAP+RAdam.
>
>
> > Try to sweep some parameters
>
> - Our protocol involves first fine-tune the baseline, then invesigate our method.
> - We added Appendix G.3 Table 11 to show the sweep for $\beta_1$ and $\beta_2$ for the Wave dataset for the baseline .
> - The results show that $\beta_1 = 0.99 \quad \beta_2 = 0.993$ achieve the best results. That is why we have used such values.
> - We added in Appendix G.4 the sweep for the RAdam baseline over the LR between $0.0001$ and $0.002$ over all datasets.
> - The added Table 12 reports the results, finding that $LR=0.001$ perform best across all datasets.
> - For this reason, we use this LR for both baseline and our method.
> - We also add an ablation in Appendix G.2 Table 11 of our ACR method parameters, which were not fine-tuned. Infact, we find that better result can be achieved with for example $\lambda=0.05$ instead of $0.1$.
>
>
>
> >Consider isolating contributions: (a) confidence-mixing only (no $\gamma_t$), (b) $\gamma_t$ only, (c) both.
>
> - We added in Appendix G.1, in Table 8 and Table 9 ablation studies for disentangling the contributions of ACR components: Variance Tracking,  Safe Fallback, Gradient Adjustment.
> - We find that combination of the components (as in ACR) leads to best performance.
>
> ---
> **Final comment**
>
> - Thanks for your efforts.
> - Please let us know if you like to see further results / discussion.
> - If we addressed some of your concers please consider showing support for our work by increasing the score torward a stronger acceptance.

---

### Official Review · Reviewer_sxXP · 2025-11-01

**Soundness:** 3
**Presentation:** 3
**Contribution:** 3
**Rating:** 4
**Confidence:** 2

**Summary:**

This paper proposes ACR to address the instability issues that arise when training PINNs with second-order optimizers. The method is based on an empirical confidence-based measure and does not assume that the gradients follow a Gaussian distribution. The authors validate the effectiveness of their approach through experiments.

**Strengths:**

Overall, the paper is convincing. The authors identify a common issue when training PINNs with second-order optimizers, namely the instability that occurs in the early stages of training due to second-order estimation. The proposed method, ACR, unlike prior work such as RAdam, does not rely on the Gaussian distribution assumption. Instead, it leverages the observed variability of second-order moment statistics to dynamically adjust the rectification strength. The authors integrate ACR into several second-order optimizers, including SOAP, AdaHessian, and Sophia, and demonstrate consistent improvements.

**Weaknesses:**

This paper relies heavily on empirical design and lacks corresponding theoretical support, such as convergence guarantees or an analysis of how ACR affects the bias and variance of the second-order moment estimates.

Although the definition of confidence appears reasonable, it introduces many additional hyperparameters. The authors do not specify how the default values of these parameters were chosen, nor do they report any ablation or sensitivity analysis regarding them, which may limit the practical applicability of the proposed method.

The paper also lacks an ablation study on the internal components of ACR.

**Questions:**

Could the authors explain how the hyperparameters were selected? Could they provide a sensitivity analysis and present more detailed ablation experiments? Does the proposed method introduce additional computational overhead?

---

> ### Author Response · Authors · 2025-11-25
>
> Thanks for your help on improving our paper.
> We have additional results to share:
>
>
> >This paper relies heavily on empirical design and lacks corresponding theoretical support,
>
> - Our added Appendix H.1 shows that ACR achieves strictly lower variance in the effective learning rate compared to RAdam. Intuitively, this happens because ACR does not fully trust the second-moment estimate $\hat v_t$ when it is still noisy.  A confidence term down-weights unstable estimates and pulls the rectification toward a conservative fallback, which shrinks fluctuations in the adaptive step size. As the estimates stabilize, the confidence increases and ACR smoothly returns to standard RAdam behavior.
> - In short: ACR reduces early-training noise by dampening unreliable variance estimates, giving more stable updates without altering RAdam’s long-run dynamics.
> - See the added Appendix H.1 for the formal definitions and derivations.
>
>
> >Could the authors explain how the hyperparameters were selected?
>
> Our protocol involves first fine-tuning the baseline, and then evaluating our proposed method.
>
> - We added Appendix G.3 (Table 11) to present the sweep over $\beta_1$ and $\beta_2$ for the Wave dataset under the baseline setting.
> - The results indicate that $\beta_1 = 0.99$ and $\beta_2 = 0.993$ yield the best performance, which is why we adopt these values.
> - In Appendix G.4, we report a learning–rate sweep for the RAdam baseline over the range 0.0001–0.002 across all datasets.
> - Table 12 shows that LR = 0.001 consistently performs best, and thus we use this learning rate for both the baseline and our method.
> - Finally, Appendix G.2 (Table 11) includes an ablation study of the ACR method’s parameters, which were not fine-tuned.
> - We observe that improved performance can be obtained with, for example, $\lambda = 0.05$ instead of the default 0.1.
>
>
>
>
> >The paper also lacks an ablation study on the internal components of ACR.
>
> - We added in Appendix G.1, in Table 8 and Table 9 ablation studies for disentangling the contributions of ACR components: Variance Tracking,  Safe Fallback, Gradient Adjustment.
> - We find that combination of the components (as in ACR) leads to best performance.
>
>
> >Could they provide a sensitivity analysis and present more detailed ablation experiments?
>
> - In Appendix G.2, we include a hyperparameter sweep for ACR over $\rho_{\text{safe}}, \tau, \lambda,$ and $\beta_s$ for the Wave dataset.
> - The results in Table 10 show that performance remains comparable across a broad range of settings, indicating that ACR is generally robust to hyperparameter choices.
> - Overall, the sweep confirms that ACR does not require fine-grained tuning to perform well.
>
>
> >Does the proposed method introduce additional computational overhead?
>
> - In Appendix G.8 we added Table 13 to summarize the wall-clock time to reach convergence for different datasets and model scales.
> - We observe that ACR incurs only a negligible overhead relative to SOAP+RAdam.
>
> ---
> **Final comment**
>
> - Please let us know if you like to see further results / discussion.
> - Thanks for your efforts.
> - If we addressed some of your concers please consider showing support for our work by increasing the score torward a stronger acceptance.

---

> > ### Comment · Reviewer_sxXP · 2025-11-27
> >
> > Thank you for the authors’ detailed response. The current version of the manuscript has shown significant improvements in both theory and experiments. However, I notice that a substantial portion of this work was completed during the rebuttal period, and the initial submission lacked sufficient completeness. Therefore, to ensure a fair evaluation, I will maintain my current score.

---

### Comment · Area_Chair_9UVm · 2025-11-27
**Request for Timely Response to Authors’ Rebuttal and Discussion**

Dear Reviewers,

I hope you are doing well. The authors have now submitted their rebuttal for the paper under your review. At this stage, your timely response is essential for ensuring a smooth discussion phase.

Could you please review the rebuttal at your earliest convenience and share your updated thoughts? If there are points that require further discussion among the reviewers, please feel free to initiate or join the conversation on the discussion thread.

Your prompt input will greatly help us maintain the review timeline. Thank you very much for your efforts and valuable contributions.

Best regards,

AC

---

### Author Response · Authors · 2025-12-03

Dear AC, we provide a summary of our rebuttal.

- Two reviewers acknowledged substantial improvements to the paper.
- The remaining two reviewers could not provide updates in time; however, all of their concerns have been addressed with additional methodological and experimental evidence.

Reviewer qKCz raised the score to 6 and confirmed that the previously raised concerns were resolved:
> “Regarding the concerns I raised, I am happy to see that almost all of them are responded.”

Reviewer sxXP noted significant improvements in both theory and experiments:
> “The current version of the manuscript has shown significant improvements in both theory and experiments.”

We added several new results:
- Appendix G.1–G.11: new experimental results.
- Appendix H.1: new theoretical results.
- Below we summarize the additional results requested by the reviewers.

---

### Ablation of Internal Components of ACR
- Appendix G.1 (Tables 8–9) includes ablations disentangling the contributions of Variance Tracking, Safe Fallback, and Gradient Adjustment.
- The full combination (ACR) achieves the best performance.
- Removing the safe fallback leads to performance degradation.

### Hyperparameter Sensitivity of ACR
- Appendix G.2 provides a sweep over $\rho_{\text{safe}}, \tau, \lambda,$ and $\beta_s$ on the Wave dataset.
- This clarifies the role of $\lambda$ as a gradient-moderation factor.
- Table 10 shows stable performance across a broad parameter range, indicating robustness.
- The sweep confirms that ACR does not require fine-grained tuning.

### Hyperparameter Search for Baseline Momentum Parameters
- Appendix G.3 (Table 11) adds a sweep over $\beta_1$ and $\beta_2$ for the Wave dataset.
- The optimal values are $\beta_1 = 0.99$ and $\beta_2 = 0.993$.
- These are therefore used in all baseline experiments.

### Hyperparameter Search for Baseline Learning Rate
- Appendix G.4 sweeps the baseline learning rate from $0.0001$ to $0.002$ across all datasets.
- Table 12 shows that $LR=0.001$ performs best consistently.
- We therefore use this LR for both the baseline and our method.

### Confidence of Second-Moment Estimates
- Appendix G.5 (Figure 8) plots the evolution of $conf_t$ and $CV_t$ during training.
- After initial instability, $conf_t$ rises toward $1$ while $CV_t$ decreases toward $0$.
- This shows that confidence evolves non-trivially early in training and may inform optimizer behavior.

### Difference Between ACR and RAdam
- Appendix G.6 (Figure 9) illustrates that $\rho_t$ in RAdam is a global scalar, while $\rho_{\text{target}}$ is per-parameter.
- Early in training, $\rho_{\text{target}}$ differs markedly from $\rho_t$, capturing parameter-specific variability.
- Its distribution narrows over time, approaching $\rho_t$, indicating that adaptivity is most important early.

### Comparing ACR and Raw Confidence Signals
- Appendix G.7 (Figure 10) compares the evolution of $r_t^{\mathrm{ACR}}$ and $conf_t$.
- KDE plots show that their distributions differ substantially until roughly 100K epochs.

### Cumulative Runtime
- Appendix G.8 (Table 13) summarizes wall-clock times to convergence for all datasets and model scales.
- ACR incurs negligible overhead relative to SOAP+RAdam.

### ACR in First-Order Settings
- Appendix G.9 (Table 14) reports L2 results comparing RAdam and ACR across datasets.
- ACR consistently improves over RAdam.

### Per-Epoch Wall-Clock
- Appendix G.10 (Table 15) provides per-epoch timings.
- SOAP-ACR adds only negligible overhead relative to RAdam.

### Testing Alternative Confidence Estimation: Bootstrap
- Appendix G.11 introduces a bootstrap-based uncertainty estimator (Algorithm 5) and compares it with ACR (Algorithm 6).
- Table 15 shows that the bootstrap variant is consistently more expensive.
- ACR achieves lower error while remaining faster.

### Variance Bound for ACR
- Appendix H.1 proves that ACR achieves strictly lower variance in the effective learning rate than RAdam.
- This occurs because ACR discounts noisy estimates of $\hat v_t$ using a confidence term that pulls rectification toward a conservative fallback.
- As estimates stabilize, confidence increases and ACR smoothly transitions to standard RAdam behavior.
- ACR therefore reduces early-training noise while preserving long-run dynamics.

---

### Meta-Review · Area_Chair_N64o · 2025-12-08

**Summary:**

This paper targets an improved method for training PINNs with second order methods. Specifically, it targets an improvements of the rectification strategy of RAdam.

Overall, the reviewers highlight that the method is intuitive and, and that experiments demonstrate consistent improvements across PDE benchmarks and optimizers.

That being said, reviewers had significant concerns initially, resulting in a uniformly negative assessment. One reviewer (qKCz) raised the score, but another explicitly did not, arguing that large parts of the results were added after the rebuttal, keeping their negative viewpoint.

Regarding the two reviewers that did not speak up, I see a more positive assessment for reviewer YZ9y, and a good chance that this reviewer would have raised the score. Reviewer EgiS on the other hand, has a more critical stance. So overall, my assessment is that this paper would have received a split score, which, due to ICLR being one of the top venues in the field, would not have been sufficient for a clear accept. My assessment of the paper is also that would profit from a stronger impact. Neither PINNs nor second order optimizers are widely used in the field, and the paper purely focuses on mild improvements in this setting. A demonstration that the method yields benefits in other contexts ("general optimizer value" as EgiS puts it) would be a good argument in favor of the method.

**Reviewer Concerns:**

The authors have submitted an extensive rebuttal,

Remaining points of criticism that remain are:
- more general settings / experiments showing general optimizer value
- attractiveness of method (more hyperparameters being introduced)
- large parts of results added post-submission

**Reviewer Scores:**

Scores: qKCz raised to 6, while sxXP kept 4.

From YZ9y, I would have expected a raise to 6, while EgiS most likely would have kept the score of 4.

---

### Decision · Program_Chairs · 2026-01-26

Reject